# Bone marrow stromal cells induce chromatin remodeling in multiple myeloma cells leading to transcriptional changes

Moritz Binder[1], Raphael E. Szalat[1,2], Srikanth Talluri[1], Mariateresa Fulciniti [1], Hervé Avet-Loiseau[3], Giovanni Parmigiani[2,4], Mehmet K. Samur [2,4] ✉ & Nikhil C. Munshi [1] ✉

The natural history of multiple myeloma is characterized by its localization to the bone marrow and its interaction with bone marrow stromal cells. The bone marrow stromal cells provide growth and survival signals, thereby promoting the development of drug resistance. Here, we show that the interaction between bone marrow stromal cells and myeloma cells (using human cell lines) induces chromatin remodeling of cis-regulatory elements and is associated with changes in the expression of genes involved in the cell migration and cytokine signaling. The expression of genes involved in these stromal inter- actions are observed in extramedullary disease in patients with myeloma and provides the rationale for survival of myeloma cells outside of the bone mar- row microenvironment. Expression of these stromal interaction genes is also observed in a subset of patients with newly diagnosed myeloma and are akin to the transcriptional program of extramedullary disease. The presence of such adverse stromal interactions in newly diagnosed myeloma is associated with accelerated disease dissemination, predicts the early development of ther- apeutic resistance, and is of independent prognostic significance. These stromal cell induced transcriptomic and epigenomic changes both predict long-term outcomes and identify therapeutic targets in the tumor micro- environment for the development of novel therapeutic approaches.

Multiple myeloma (MM) is a malignancy of plasma cells with marked genomic heterogeneity and highly variable clinical outcomes[1–3]. With the advent of novel therapeutic agents and drug combinations, response rates and survival outcomes have improved over time, albeit at the cost of perpetual treatment in most patients[4,5]. Despite these therapeutic advances, the natural history of the disease is characterized by emerging therapeutic resistance and frequent relapses, eventually culminating in refractory disease. Particularly, the development of extramedullary disease (EMD) at the time of relapse represents a turning point in a patient's trajectory as it

indicates not only acquired therapeutic resistance but also the acquired ability of a plasma cell clone to survive outside its natural bone marrow environment[6,7]. This secondary dissemination of dis- ease is markedly different in biology and prognosis from primary plasma cell leukemia[8–10]. In studies examining the cause of death of patients with MM, the vast majority of deaths can be attributed to either disease progression or immediate complications of myeloma- directed therapy, suggesting that survival outcomes are largely determined by the development of therapeutic resistance and ensuing disease progression[11,12].

[1]Department of Medical Oncology, Dana Farber Cancer Institute, Boston, MA, USA. [2]Department of Data Science, Dana Farber Cancer Institute, Boston, MA, USA. [3]University Cancer Center of Toulouse, Institut National de la Santé, Toulouse, France. [4]Department of Biostatistics, Harvard T.H. Chan School of Public Health, Boston, MA, USA. ✉e-mail: mehmet_samur@dfci.harvard.edu; nikhil_munshi@dfci.harvard.edu

The mechanisms by which MM cells home to the BM and adhere to BM stromal cells (BMSCs) and extracellular matrix proteins have been extensively investigated to understand their effects on MM cell growth and survival [13,14]. Several studies have investigated the molecular impact of the BMSC-MM cell adhesion process. The interaction between MM cells and BMSCs through adhesion has been tied to drug-induced apoptosis and NF-kB-dependent transcription and secretion of IL-6, which controls MM cell growth and survival [15,16]. The biological role of ERK / MAPK, JAK / STAT, and PI3-K / Akt pathways in growth, survival, and drug resistance of MM cells has also been demonstrated [17]. Transcriptional changes in MM cells in response to BMSCs have been associated with resistance to corticosteroid therapy, but the implications of stromal interactions with current anti-myeloma therapies remain unknown [18]. Although the underlying regulatory mechanisms driving the MM-BMSC interactions and their sequelae remain mostly elusive, a small

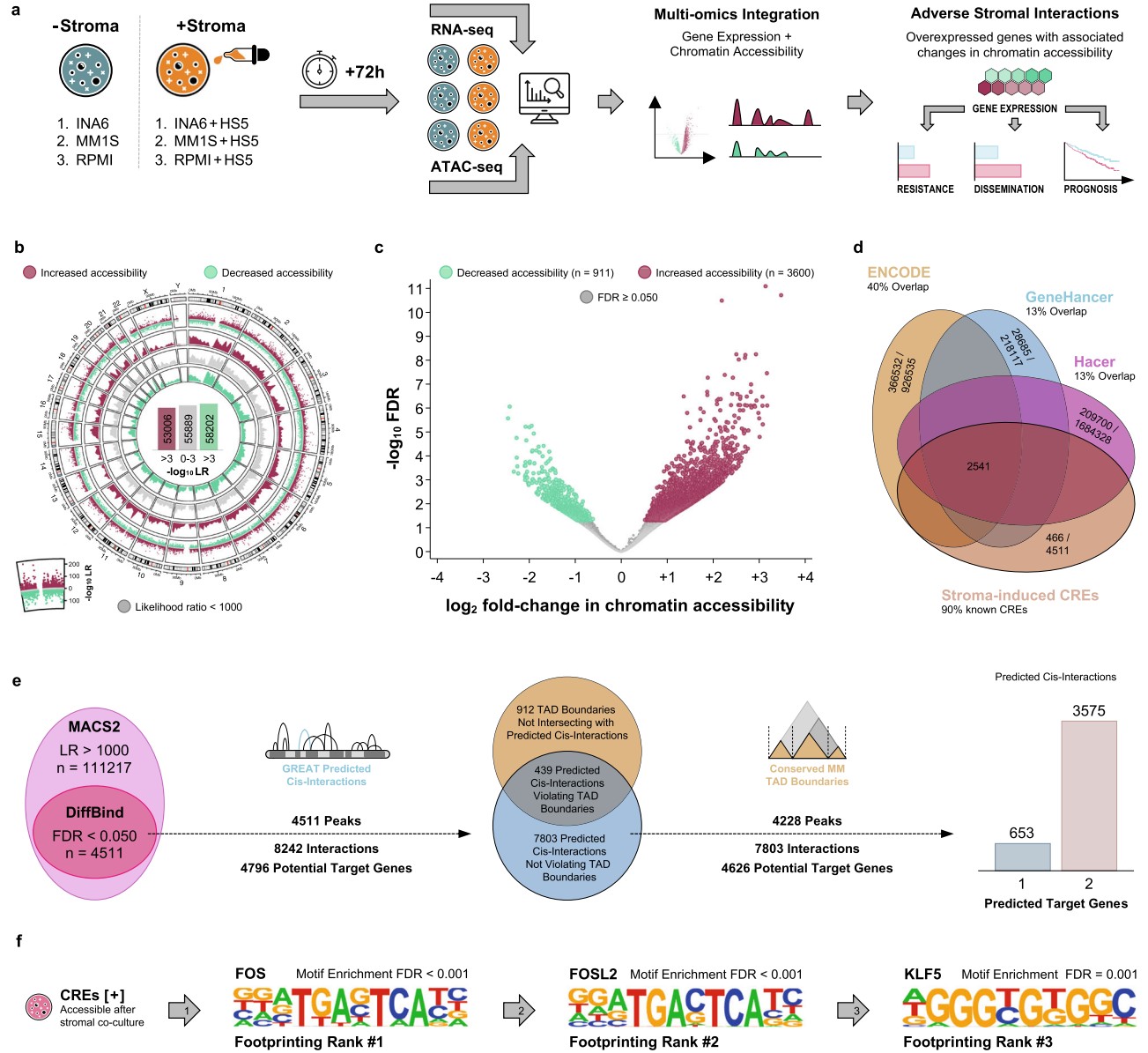

**Fig. 1 | The interaction of bone marrow stroma and myeloma cells induces chromatin remodeling with altered accessibility in known cis-regulatory elements (CREs) in three myeloma cell lines. a** Experimental design to identify changes in chromatin accessibility and gene expression in three myeloma cell lines after 72 h of co-culture with bone marrow stromal cells. **b** Circular scatter plot showing regions with increased and decreased chromatin accessibility after stromal co-culture across the entire genome. There was widespread chromatin remodeling involving a large number of genomic regions with both increased and decreased chromatin accessibility. We used this approach as a first step to select 111217 candidate regions with altered chromatin accessibility after stromal co-culture. **c** Volcano plot showing differentially accessible genomic regions (FDR < 0.050). These 4511 differentially accessible regions represent a subset of the initially identified 111217 candidate regions. **d** Venn diagram demonstrating the overlap between the 4511 differentially accessible regions and three databases of annotated human cis-regulatory elements. **e** Euler diagrams and bar graph depicting the filtering and prediction strategy employed to associate the 4511 differentially accessible regions with potential target genes within previously described conserved myeloma-specific topologically associating domains (TADs)[48]. After filtering, there were 4288 differentially accessible regions predicted to interact with 4626 potential target genes in close proximity. The 4288 differentially accessible regions were predicted to interact with either one or two potential target genes after applying all filtering criteria. **f** Position-specific weight matrices showing the top 3 transcription factors expected to bind in the genomic regions accessible after stromal co-culture (selected by transcription factor footprinting)[53]. Motif enrichment analyses are also shown. Among the top enriched transcription factors were FOS, FOSL2, and KLF5. Source data are provided as a Source Data file.

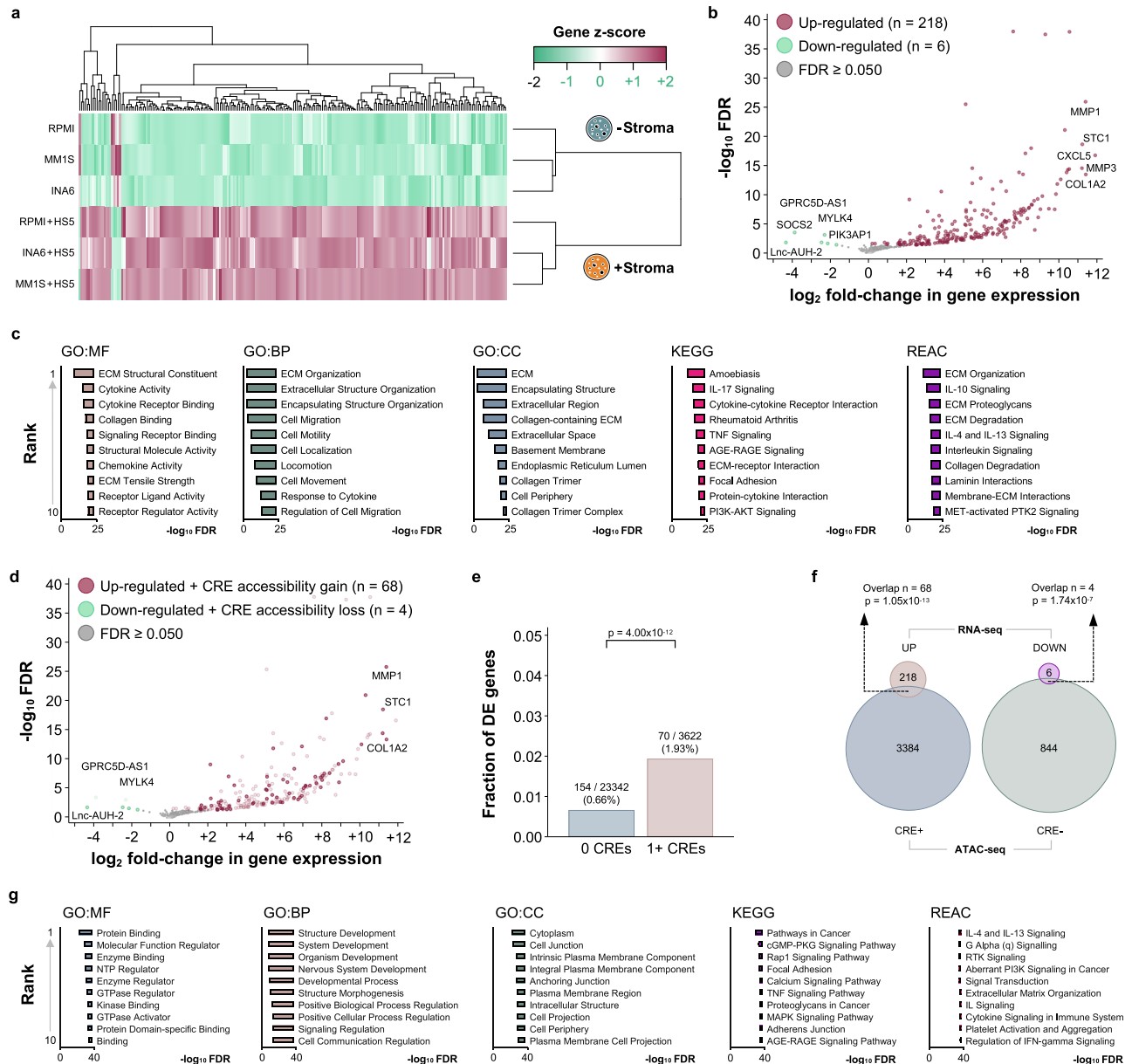

**Fig. 2 | The interaction of bone marrow stroma and myeloma cells induces up-regulation of genes involved in cell migration and cytokine signaling in human myeloma cell lines.** This up-regulation is supported by the presence of de novo chromatin accessibility in associated cis-regulatory elements. **a** Heatmap (unsupervised hierarchical clustering) showing separation of the experimental conditions based on gene expression (presence *versus* absence of stromal co-culture). **b** Volcano plot demonstrating the up-regulation of genes in myeloma cells due to interactions with bone marrow stroma. Genes coding for interleukins, chemokine ligands, and matrix metalloproteinases were among the top up-regulated genes (top 5 genes in each direction labeled). **c** Bar graphs listing the top 10 functional annotation terms for the 224 differentially expressed genes (GO = Gene Ontology, MF = Molecular Function, BP = Biological Process, CC = Cellular Compartment, KEGG = Kyoto Encyclopedia of Genes and Genomes Pathway, REAC = Reactome Pathway). These top 10 terms included cytokine and interleukin signaling as well as cell migration and remodeling of the extracellular matrix. **d** Volcano plot demonstrating differential expression of genes with concomitant differential chromatin accessibility in associated CREs. Genes coding for interleukins, chemokine ligands, and matrix metalloproteinases remained among the top up-regulated genes. The top 3 differentially expressed genes with associated differentially accessible CREs are labeled. **e** Bar graph showing the overrepresentation (Fisher's exact test) of up-regulated genes among the genes with at least one CRE with altered accessibility nearby (i.e. genes close to CREs with altered chromatin accessibility after stromal co-culture are more likely to be up-regulated after stromal co-culture compared to genes without such CREs nearby). **f** Venn diagrams demonstrating that the observed association between differential expression and differential chromatin accessibility in corresponding CREs is unlikely to have arisen by chance (hypergeometric test). **g** Bar graphs listing the top 10 functional annotation terms for the 4626 predicted target genes predicted to be involved in cis-interactions. These top 10 terms again included interleukin and cytokine signaling as well as several cancer-related pathways. Source data are provided as a Source Data file.

number of studies have implicated epigenetic regulatory mechanisms. Promoter methylation, histone modifications, and chromatin accessibility have been found to be involved in the MM-BMSC interactions and regulation of specific genes involved in cell signaling, osteoblast suppression, and therapeutic resistance[19–24]. The

causal relationships between the expression of specific genes and extramedullary dissemination and therapeutic resistance raises the question whether the underlying regulatory mechanisms governing the expression of these genes could be exploited for therapeutic benefit.

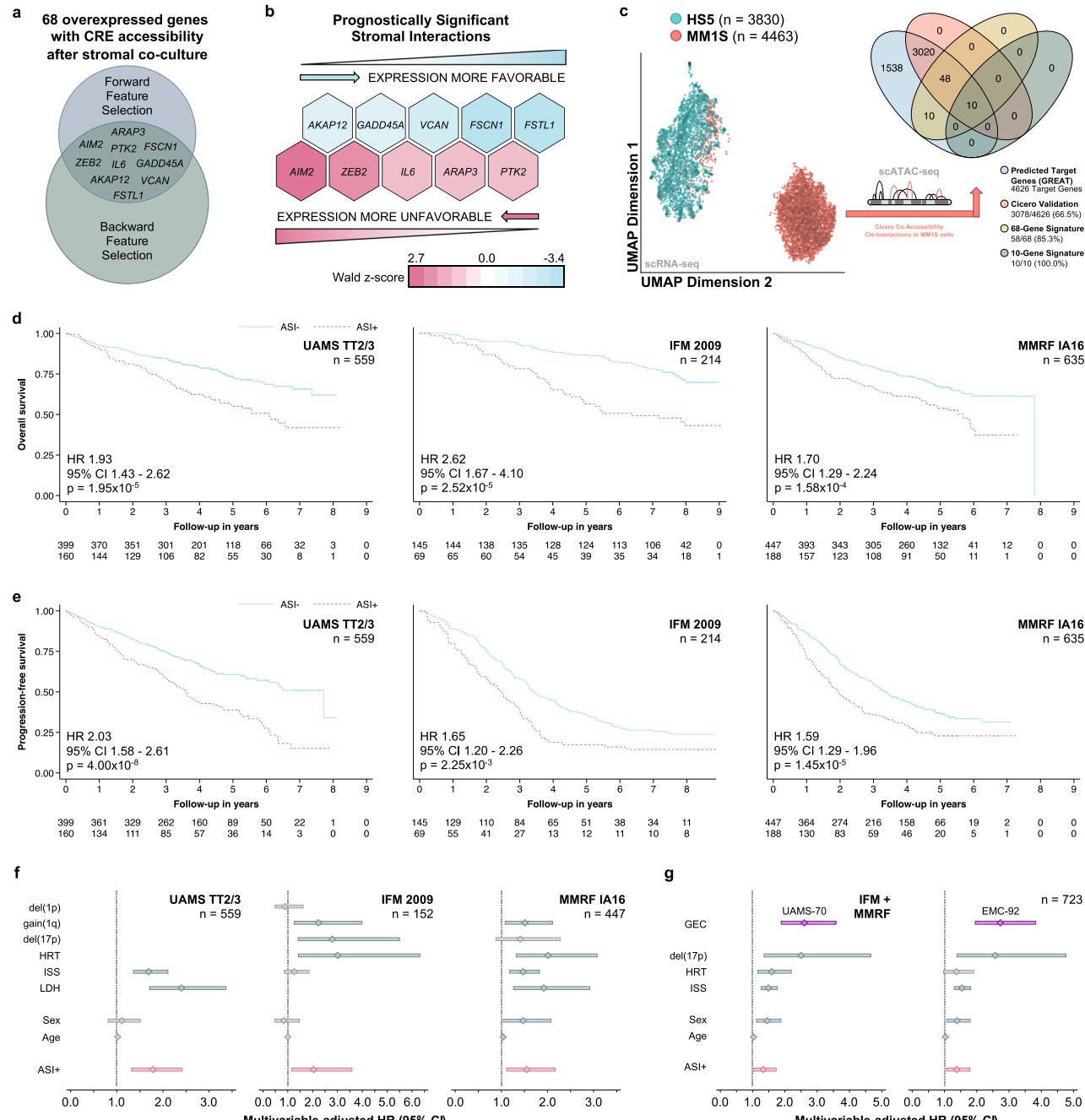

**Fig. 3 | The expression of stromal interaction genes is of independent prognostic significance in myeloma and defines a high-risk subgroup of patients with distinct transcriptomic features. a** Venn diagram showing the subset of genes independently associated with progression-free survival in patients with newly diagnosed myeloma (UAMS TT2/3, $n = 559$, GSE24080). The consensus of forward and backward feature selection in proportional hazards regression models identified 10 genes that were selected for further study based on their independent prognostic significance. **b** Heatmap showing the direction and effect sizes for the 10 genes independently associated with progression-free survival in UAMS TT2/3. The expression of half of the genes was associated with increased progression-free survival, the expression of the other half with decreased progression-free survival. The color gradient reflects the Wald z-score of the regression coefficient for each gene in the multivariable-adjusted proportional hazards regression model (a normalized measure of the effect size). **c** Scatter plot (left) showing the separation of HS5 (turquoise) and MM1S (orange) cells after co-culture in two-dimensional UMAP space based on their single-cell gene expression profiles. Euler diagram (right)

showing the validation the predicted cis-interactions in MM1S cells after co-culture with HS5 using single-cell co-accessibility analysis. **d** Kaplan-Meier plots showing the association between the presence of adverse stromal interactions (ASI +) and overall survival (Wald test). **e** Kaplan-Meier plots showing the association between the presence of adverse stromal interactions (ASI +) and progression-free survival (Wald test). **f** Forest plots demonstrating the independent prognostic significance (overall survival) of the presence of adverse stromal interactions (ASI +) in the derivation cohort and the two validation cohorts, when adjusting for established clinical, laboratory, and cytogenetic high-risk markers. The center marker represents the hazard ratio (HR) and the bars the corresponding 95% confidence interval. **g** Forest plots demonstrating the independent prognostic significance (overall survival) of the presence of adverse stromal interactions (ASI +) in the combined validation cohorts when adjusting for the established clinical, laboratory, as well as the UAMS-70 and EMC-92 high-risk gene expression classifier, respectively. Center marker and bars same as for (**f**). Source data are provided as a Source Data file.

These observations suggest that there are specific transcriptional programs responsible for therapeutic resistance and EMD dissemination. However, it remains unclear if the presence of structural variation within the MM cells can fully explain these prognostically important disease phenotypes. One alternative explanation to a unifying regulatory mechanism determined by structural variation within the MM cells are dynamic epigenetic changes incited by interactions with the tumor microenvironment. The latter is conceivable given the intricate bidirectional relationship between MM cells and BMSCs[14,25]. Stromal cells have been demonstrated to protect MM cells against cytotoxic treatment and immune related injury by altering gene expression in the MM cells and suppressing T-cell-mediated anti-tumor immune responses[26,27].

Here, we investigate the chromatin remodeling and associated changes in gene expression in the MM cells following MM-BMSC interactions and show their implications for the development of therapeutic resistance, EMD dissemination, and patient outcomes.

## Results

### BMSCs induce chromatin remodeling of cis-regulatory elements in MM cell lines

To identify chromatin accessibility and consequent gene expression changes induced in MM cells by their interaction with BMSCs, we performed DNA transposase-accessibility assays (ATAC-seq) and whole transcriptome sequencing (RNA-seq) in three human MM cell lines after 72 h of co-culture with and without a human stromal cell line (Fig. 1a). We observed widespread chromatin remodeling in the MM cells following co-culture with BMSCs: There were both increases (formation of euchromatin) and decreases in chromatin accessibility (formation of heterochromatin, Fig. 1b, Supplementary Fig. 1a). We identified 4511 regions with altered accessibility among 111217 candidate regions by performing differential accessibility analysis (Fig. 1c). Intersection with public databases of human cis-regulatory elements (CREs) confirmed that the vast majority of these 4511 genomic regions represented known CREs (Fig. 1d). However, only few of these regions were located in conserved MM super-enhancer regions (8 of 4511, 0.18%)[28]. We identified 4626 potential target genes associated with 4288 CREs using proximity on the linear genome and localization within conserved MM topologically associating domains as selection criteria (Fig. 1e, Supplementary Data 1). The 4288 CREs associated with potential target genes were predicted to bind a number of transcription factors by transcription factor footprinting, including several members of the FOS, KLF, and IRF families (Fig. 1f).

### Chromatin accessibility of cis-regulatory elements is associated with increased transcription of their potential target genes in MM cell lines

As the vast majority of these elements are known CREs, we investigated whether their accessibility is associated with the transcription of their predicted target genes. Unsupervised hierarchical clustering separated the experimental conditions and revealed marked differences in gene expression between stroma-exposed and -unexposed MM cell lines (Fig. 2a). We first examined the gene expression profiles of the three cell lines before and after stromal co-culture using pairwise correlation (Supplementary Fig. 1b-d). Gene expression before and after stromal co-culture are highly correlated (r > 0.90 and $p < 0.001$ for all three cell lines). Next, we categorized transcripts as down-regulated (decrease in normalized transcript count), stable (no change in normalized transcript count), and up-regulated (increase in normalized transcript count) after stromal co-culture for each cell line. There was strong agreement between the three cells lines in terms of the transcriptional changes in response to stromal co-culture (inter-rater agreement >0.70 and $p < 0.001$ for all three cell lines). Differential gene expression analysis between the two experimental conditions demonstrated a predominant up-regulation of transcriptional activity

in the MM cells after exposure to bone marrow stroma cells (218 gene up-regulated, 6 genes down-regulated, Fig. 2b, Supplementary Data 2). The up-regulated genes included several cytokines, chemokines, and matrix metalloproteinases. Functional annotation revealed the activation of pathways related to extracellular matrix organization and cell migration in the MM cells (Fig. 2c, Supplementary Data 2). The up-regulation of these genes was specific to stromal co-culture and not a baseline property of the employed MM cell lines. Among 70 commonly used MM cell lines at baseline (without stromal co-culture), only two show increased expression in >50% of the dysregulated genes after stroma co-culture. Likewise, we did not observe an enrichment in the expression of the 68 dysregulated genes in MM cell lines with high-risk IGH translocations (Enrichment Score −0.57, $p = 0.644$, Supplementary Fig. 2a-b). First, we observed that there was significant overlap between the differentially expressed genes and the predicted target genes of CREs with altered chromatin accessibility (Fig. 2d). Furthermore, genes with altered chromatin accessibility in their associated CREs were significantly overrepresented among the differentially expressed genes (Fig. 2e) and the observed overlap was highly unlikely to have arisen by chance (Fig. 2f). Additionally, the median expression of genes with an associated differentially accessible CRE was higher compared to genes without an associated CRE, without there being a dosage effect associated with CREs (Supplementary Fig. 2c). The majority of the CREs were located outside promoter regions and within 500 kilobases of the transcription start site (Supplementary Fig. 2d). The effect of these CREs on gene expression decreased as the distance from their predicted target genes increased (Supplementary Fig. 2e). Functional annotation of the predicted target genes again revealed interleukin and cytokine signaling as well as several cancer-related pathways, extracellular matrix organization, and cell migration (Fig. 2g, Supplementary Data 3).

### MM / BMSC interaction-induced expression of genes has independent prognostic significance in patients with newly diagnosed MM

Next, we evaluated whether the expression of genes governed by stromal interactions is of prognostic significance in patients with MM. We used the 68 genes with concordant CRE accessibility and expression (overexpression and de novo accessibility in associated CREs, Fig. 2d) to test the association between their expression and progression-free survival in 559 newly diagnosed uniformly treated patients with MM. Employing automated feature selection methods, we identified 10 of the 68 genes to be independently associated with survival in the derivation cohort (Fig. 3a, Supplementary Table 1–3). Half the genes were associated with favorable survival while the other half were associated with decreased survival (Fig. 3b). We further generated single-cell multi-omics data (scRNA and scATAC sequencing from the same cells) to validate the predicted cis-interactions between enhancer and predicted target genes induced by MM stroma interactions. This experiment confirmed all ten interactions identified from bulk sequencing data (Fig. 3c, Supplementary Fig. 3a). While the expression of these 10 genes was of independent prognostic significance, 90% of them had not previously been implicated in high-risk disease (Supplementary Fig. 3b). To investigate the combined effects of the expression of both protective and harmful stromal interactions, we devised a simple summary measure of the expression of these 10 genes and dichotomized the obtained values (please see *Methods* for details, Supplementary Fig. 3c−f, Supplementary Table 4) to designate a group of patients with MM cells not having adverse stromal interactions (ASI-) signature and a group with presence of ASI (ASI + ). Using this ASI classifier, we validated the adverse prognostic impact of ASI+ in two independent patient populations Fig. 3d, e). Importantly, the prognostic significance of ASI was independent of the established high-risk disease markers (Fig. 3f, Supplementary Table 5–7). In addition to the minimal or absent overlap between the identified stromal

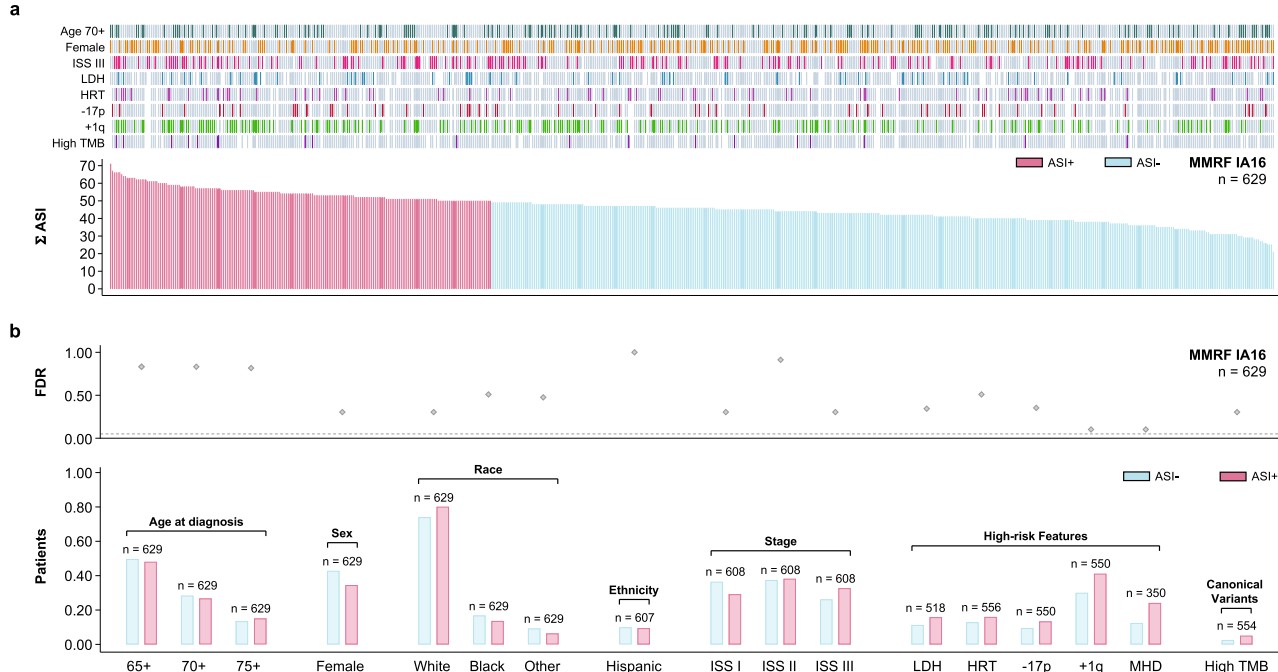

**Fig. 4 | The presence of adverse stromal interactions (ASI + ) is a distinct high-risk patient subgroup and occurs in patients with newly diagnosed myeloma.** **a** Bar graphs and heat map showing the distribution of high-risk characteristics (ISS International Staging System, LDH Elevated lactate dehydrogenase, HRT Presence of a high-risk IGH translocation, −17p = Del(17p), +1q = Gain(1q), TMB high tumor mutational burden) among 623 patients with newly diagnosed myeloma ranked by the extent of adverse stromal interactions (Σ ASI). **b** Bar graphs showing the distribution of demographic characteristics (MHD = Multi-hit disease; double- or triple-hit) of patients stratified by ASI + . The scatter plot indicates the statistical significance for each two-way comparison (ASI+ versus ASI−) after adjusting for multiple comparisons (Fisher's exact test).

interaction genes and existing transcriptome-based high-risk classifiers, the prognostic implications of the former were independent of the latter (Fig. 3g, Supplementary Table 8-9).

**Adverse stromal interactions define a hitherto unrecognized high-risk subtype of MM and are associated with therapeutic resistance in patients with newly diagnosed MM**

Since the presence of ASI was of independent prognostic significance, we investigated their potential co-segregation with the established high-risk disease markers in patients with MM. The presence of traditional high-risk disease markers was balanced across the extent of ASI in validation cohort II (MMRF, $n = 623$, Fig. 4a). Likewise, the distribution of demographic, laboratory, cytogenetic, and genomic characteristics was similar between patients with adverse stromal interactions and those without (Fig. 4b). These results were consistent with validation cohort I (IFM, $n = 214$): Age at diagnosis ($p = 0.658$), sex ($p = 0.700$), and International Staging System stage ($p = 0.962$). Furthermore, the mutational spectrum of patients with ASI was not different from those without (Fig. 5a, b). Since in vitro stromal interactions are known to impact the development of therapeutic resistance, we further evaluated whether ASI are associated with response to therapy. We defined early therapeutic resistance as stable or progressive disease (by IMWG uniform response criteria) three months after initiation of first-line treatment. The choice of an early response endpoint was motivated by the high overall response rates (ORR) and resulting insufficient sample sizes in both MMRF IA16 (ORR 95%, 959 of 1014 patients) and IFM 2009 (ORR 98%, 684 of 700 patients) when attempting to model best overall response to first-line treatment. The presence of ASI was associated with early therapeutic resistance in general and with resistance to VRd (bortezomib, lenalidomide, and dexamethasone) in particular (Fig. 5c). The presence of ASI remained predictive after adjusting for the established high-risk disease markers, confirming that none of them were predictive of

treatment response (Fig. 5d, Supplementary Table 10–11). The association with early therapeutic resistance to VRd translated into decreased progression-free survival both among patients receiving VRd in MMRF (Fig. 5e) and patients in IFM 2009 (Fig. 3e, Supplementary Tables 12–13, all patients receiving VRd per protocol). In addition to being independent from the established high-risk disease markers, the association between ASI and progression-free survival remained consistent in both patient populations after limiting the analysis to patients who achieved a response to first-line treatment (Supplementary Tables 14–15). This observation supports the notion that the prognostic significance of ASI is not exclusively due to by their ability to identify non-responders.

**Stromal interaction related gene expression changes recapitulate the transcriptional program of extramedullary disease and lead to accelerated disease dissemination in patients with MM**

To test whether MM cells that survive outside their natural bone marrow environment sustain the expression of such stromal-interaction-induced genes, we examined the expression of the 68 genes with concordant expression and CRE accessibility (over-expression and de novo accessibility in associated CREs, Fig. 2d) in EMD manifestations. We compared MM cells from malignant effusions and circulating plasma cells (CPCs) to their bone marrow counterparts. We observed concordant expression of the 68 stromal interaction genes in both EMD (Fig. 6a) and CPCs (Fig. 6b). Conversely, patients with ASI experienced accelerated disease dissemination in the form of increased rates of detectable CPCs at the time of diagnosis (Fig. 6c). Moreover, a greater extent of ASI was associated with higher numbers of detectable CPCs (Fig. 6d). This translated into an increased prevalence of disseminated bone disease at the time of diagnosis (Fig. 6e) and an increased incidence of progressive bone and soft tissue disease during follow-up (Fig. 6f, Supplementary Tables 16–17).

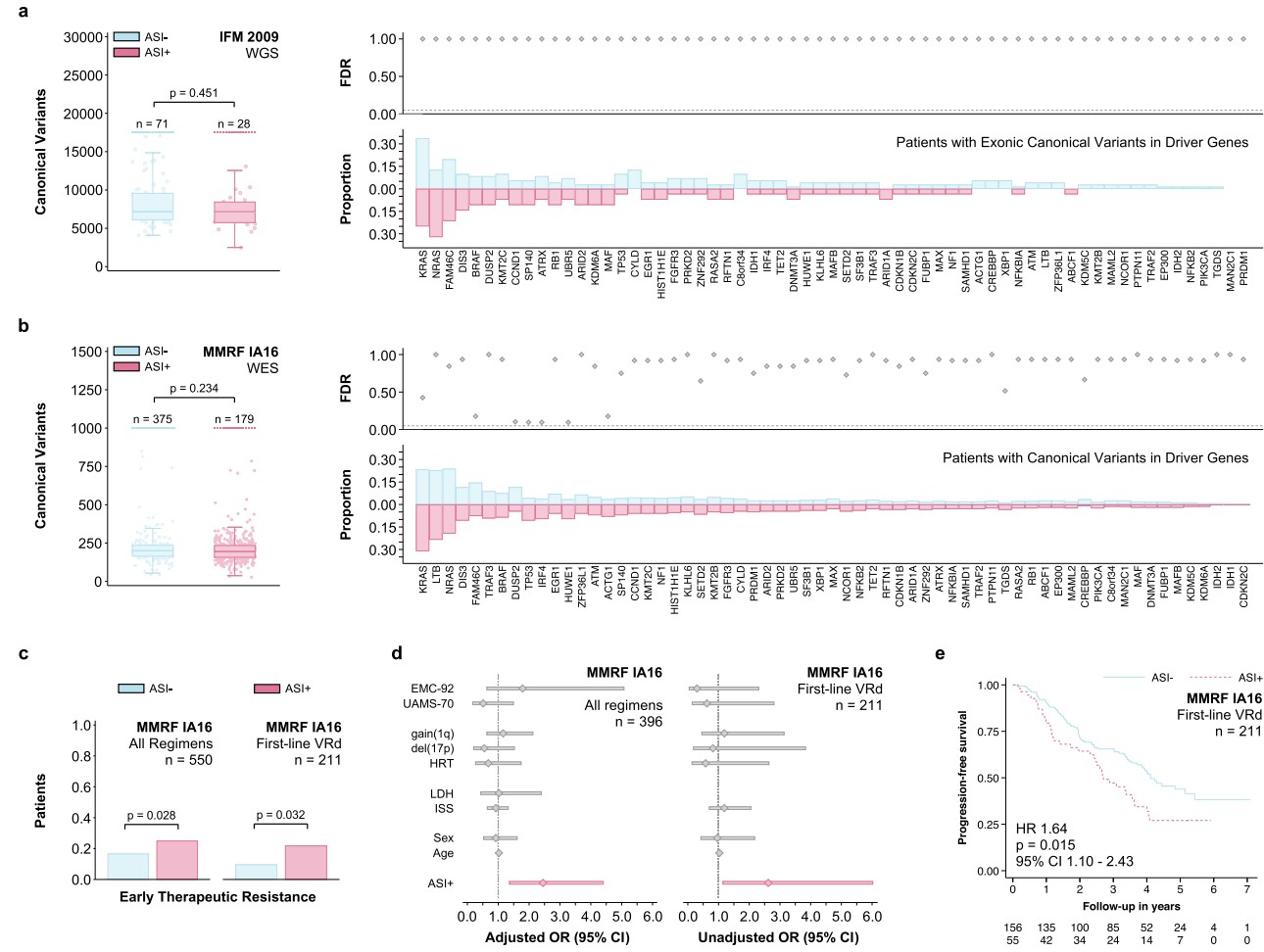

**Fig. 5 | The presence of adverse stromal interactions (ASI + ) is not associated with a specific genotype and is predictive of therapeutic resistance to first-line therapy. a** Box and strip plots showing the number of genome-wide canonical variants in the first validation cohort stratified by the presence of ASI+ (Mann–Whitney-U test). Bar graphs showing the proportion of patients with exonic canonical variants in 63 recurrently mutated myeloma driver genes[3]. The scatter plot indicates the statistical significance for each two-way comparison (ASI+ versus ASI−) after adjusting for multiple comparisons (Fisher's exact test). **b** Box and strip plots showing the number of exome-wide canonical variants stratified in second validation cohort by the presence of ASI+ (same analysis and data presentation as in **a**). **c** Bar graphs showing the frequency of early therapeutic resistance (Progressive Disease or Stable Disease by IMWG unified response criteria[72] three months after initiation of first-line treatment; Fisher's exact test) among patients in the second validation cohort receiving either any first-line regimen or VRd (bortezomib, lenalidomide, and dexamethasone). **d** Forest plots demonstrating the predictive

power (early therapeutic resistance) of the presence of adverse stromal interactions (ASI + ) in the second validation cohort. For all first-line regimens the sample size allowed multivariable-adjustment for the established clinical, laboratory, and cytogenetic high-risk markers. For first-line VRd, the limited sample size discouraged us from extensive modeling (unadjusted odds ratios given). The center marker represents the odds ratio (OR) and the bars the corresponding 95% confidence interval. **e** Kaplan-Meier plot showing the association of ASI+ with progression-free survival among patients in the second validation cohort receiving first-line VRd (Wald test). This association remained consistent after adjusting for age, sex, ISS, HRT, and −17p (HR 1.59, 95% CI 1.01–2.51, p = 0.044, n = 176). There was no evidence for a violation of the proportional hazards assumption (p = 0.327)[71]. Data are presented as standard Tukey boxplots (with the box encompassing Q1 to Q3, the median denoted as a central horizontal line in the box, and the whiskers covering the data within ±1.5 IQR in **c**, **d**). Source data are provided as a Source Data file.

## Discussion

We observed widespread chromatin remodeling along with a predominant up-regulation of genes involved in cytokine signaling and cell migration using transcriptomic and epigenomic data of MM cell lines after exposure to BMSCs. Interestingly, the vast majority of the identified regions of altered chromatin accessibility were CREs predicted to govern the expression of genes involved in cytokine signaling, cell migration, and the regulation of apoptosis. Furthermore, the accessibility of these CREs correlated with the transcription of their predicted target genes. These observations support the notion that altered chromatin accessibility may serve as a plausible explanation for the altered expression of genes involved in clinically important disease phenotypes[21]. Our multi-omics integration approach is different from that of *Dziadowicz* et al. in that we make a direct connection between

altered chromatin accessibility and gene expression in our model system, rather than connecting altered chromatin accessibility to a list of previously established high-risk genes in MM[22]. The majority of the identified CREs were located outside promoter regions suggesting that they may represent enhancers. The spectrum of transcription factors predicted to bind to these enhancers revealed several interesting candidates including several AP-1 family transcription factors. While JUN expression has been linked to MM cell proliferation and drug resistance in model systems, FOS expression has been implicated in clonal evolution and disease progression in patients with MM[29,30]. Enhancers governing the expression of genes involved in the development of therapeutic resistance and metastasis are of considerable interest as potential drug targets for emerging classes of small molecules with lineage- and context-specific therapeutic effects such as BET

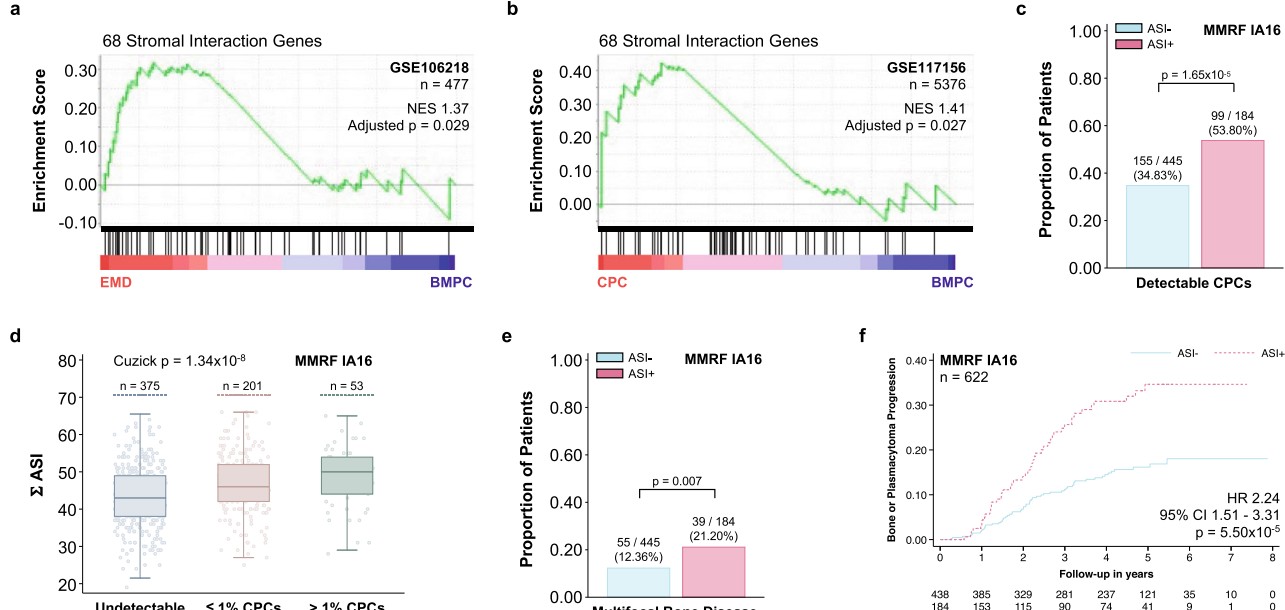

**Fig. 6 | Adverse stromal interactions cause a transcriptomic program that is akin to the gene expression profile of extramedullary disease (EMD) and are associated with accelerated disease dissemination. a** Gene Set Enrichment Analysis demonstrating the concordant expression of the 68 overexpressed genes with de novo chromatin accessibility in their associated cis-regulatory elements (see **b**) in 303 malignant extramedullary plasma cells (obtained from malignant ascites and pleural effusions) compared to 177 malignant bone marrow plasma cells (BMPCs) from the same patient population (GSE106218; permutation test, *p* value adjusted for multiple comparisons using family-wise error correction). **b** Gene Set Enrichment Analysis demonstrating the concordant expression of the same 68 genes in 2688 circulating plasma cells (CPCs) compared to 2688 BMPCs from the same population (permutation test, *p* value adjusted for multiple comparisons using family-wise error correction). **c** Bar graphs showing the distribution of patients with detectable CPCs by flow cytometry at the time of diagnosis in the

stratified by ASI+ (Fisher's exact test). **d** Box and strip plots demonstrating an increase in detectable CPCs with a greater extent of adverse stromal interactions (Σ ASI; Mann–Whitney U test). **e** Bar graphs showing the distribution of patients with imaging findings of multiple disseminated bone lesions at the time of diagnosis stratified by ASI+ (Fisher's exact test). **f** Kaplan-Meier plot showing the association of the presence of adverse stromal interactions (ASI + ) with the development of new bone lesions or soft tissue plasmacytomas during follow-up (or increase in the size of existing bone lesions or soft tissue plasmacytomas; Wald test). This association remained consistent after adjusting for age, sex, ISS, LDH, HRT, −17p, and +1q (HR 2.18, 95% CI 1.35–3.51, *p* = 0.001, *n* = 442). There was no evidence for violations of the proportional hazards assumption (*p* = 0.212)[71]. Data are presented as standard Tukey boxplots (with the box encompassing Q1 to Q3, the median denoted as a central horizontal line in the box, and the whiskers covering the data within ±1.5 IQR in **d**).

bromodomain inhibitors[28,31]. It is important to note that a stromal cell line model system was used to provide a uniform interaction and the observed changes in gene expression reflect a specific response to stromal exposure. Future studies evaluating the impact of bone marrow microenvironment heterogeneity on MM cells at single-cell resolution will be required to further extend our findings[18].

Plasma cell disorders with EMD manifestations include plasma cell leukemia, MM with CPCs, and with soft tissue plasmacytomas. All of these disease states are associated with drug resistance and poor survival outcomes compared to their counterparts without EMD manifestations[6,32,33]. The presence of EMD has raised a number of questions in the past as MM cells are known to depend on their interaction with BM stromal elements. Moreover, these interactions provide growth and survival signals to MM cells and also lead to development of drug resistance. Our results suggest that the MM cells which are able to survive as EMD outside of the BM microenvironment may have acquired the same transcriptional characteristics as induced by this interaction and hence may be able to grow out independently. Examining the expression of the stromal interaction genes in extramedullary plasma cells, we demonstrated that the gene expression observed in patients with adverse stromal interactions is similar to the transcriptional program of EMD. We did observe gene expression akin to EMD in MM cells from a considerable number of patients with newly diagnosed MM, supporting the hypothesis that there is a specific transcriptional program that can be activated even in the absence of the bone marrow microenvironment (EMD-like phenotype). This suggests that the MM cells of a subset of patients have already acquired an

adverse EMD-like transcriptional program at the time of diagnosis, long before they develop multi-drug resistance and symptomatic EMD manifestations. We did find evidence of this EMD-like phenotype in the form of increased numbers of CPCs, more widespread skeletal morbidity, and accelerated bone and soft tissue disease dissemination among patients with adverse stromal interactions.

Integrating gene expression and chromatin accessibility in MM cell lines, we identified 68 genes characterized by transcriptional up-regulation in response to stromal exposure supported by euchromatin formation in nearby CREs. We selected 10 prognostically significant MM-stromal interaction genes for further study, 6 of which (*AKAP12*, *ARAP3*, *FSTL1*, *GADD45A*, *IL6*, *VCAN*) had previously been implicated in stromal interactions either by differential expression or accessibility[18,22]. Using a simple summary measure of the expression of these 10 prognostically significant stromal interaction genes, we demonstrated that the presence of stromal interactions in patients with newly diagnosed MM (increased expression of adverse genes and / or decreased expression of protective genes) is a prognostic factor with impact on survival outcomes independent of the established high-risk disease markers including other transcriptome-based classifiers. Similarly, stromal interactions were observed across a wide spectrum of patients with MM and did not co-segregate with the established high-risk features, demographic characteristics, the overall burden of structural variation, or structural variation in the coding regions of putative driver genes. We also observed that the expression these genes was strongly associated with survival outcomes in three separate patient populations with newly diagnosed MM, receiving

contemporary anti-myeloma therapies. The function of four of these genes has already been studied by us or others (*IL6, PTK2, AIM2* and *GADD45A*) and two additional genes had previously been implicated in MM biology or prognosis (*AKAP12* and *VCAN*)[18,34–37]. As exemplified by these genes, adverse stromal interactions can induce the expression of genes involved in tumor proliferation and may be a plausible explanation for the development of EMD in MM. Further in-depth molecular and cell signaling investigations will be required to delineate functional impact of these genes in MM.

While the presence of adverse stromal interactions identified yet another high-risk subtype of MM, its implications for therapeutic target discovery may be unique. The property that sets adverse stromal interactions apart from all other established high-risk disease markers is their ability to predict response to treatment. We observed a more than two-fold increase in the odds of early therapeutic resistance in patients receiving the current standard of care (VRd) and found similar predictive abilities when generalizing to all first-line regimens. Interestingly, the stromal interaction related changes in expression we observed neither involved any of the genes reported by *Ubels* et al. (predictive of response to proteasome inhibitors) nor any of the genes reported by Bhutani et al. (predictive of response to immunomodulators) even though VRd is the archetypal regimen containing both drug classes[38,39]. The adverse stromal interactions identified here predicted response to contemporary anti-myeloma therapies, which is different from previously identified transcriptomic and epigenomic classifiers[18,22]. Once the data from ongoing clinical trials and the use of novel cellular therapies in clinical practice become mature, it will be important to investigate the role of stromal interactions in the development of resistance to CAR T-cell and bispecific antibody therapies[27]. Going back to the notion that survival outcomes in MM are largely determined by the development of therapeutic resistance and EMD manifestations, identifying a group of genes associated with both raises the possibility that these genes are not merely high-risk disease markers but may be causally related to these outcomes so crucially important to patients with the disease.

In summary, we identified a transcriptional program in MM cell lines that is induced by bone marrow stroma and associated with chromatin remodeling of CREs. We discovered that the expression of such stroma-induced genes recapitulates the transcriptional program of EMD in patients and represents a hitherto unrecognized high-risk subtype of MM associated with early development of therapeutic resistance, accelerated disease dissemination, and increased morbidity and mortality. Identifying novel transcriptomic and epigenomic targets in the tumor microenvironment is of considerable interest for the development of therapeutic approaches to prevent emerging drug resistance and to further improve outcomes of patients with MM.

## Methods

This research complies with all relevant ethical regulations of the participating institutions that approved the study protocol.

### Cell culture experiments

The human MM cell lines MM.1 S (ATCC cat # CRL-2974), RPMI-8226 (ATCC cat # CCL-155), INA-6 (DSMZ cat # ACC-862) representative of different genetic backgrounds, and the human stromal cell line HS5 (ATCC cat # CRL-3611) were cultured in RPMI-1640 medium supplemented with complete medium (10% fetal bovine serum, 100 units/mL penicillin, 100 µg/mL streptomycin, and 2 mM L-glutamine) at 37 °C and 5% CO2. Recombinant human IL-6 at a concentration of 1 ng/mL (R&D Systems, Minneapolis, United States) was added to INA-6 culture medium. The MM.1 S, RPMI-8226, and HS-5 cells were obtained from ATCC and authenticated using ATCC STR profiling protocols. INA-6 was obtained from DMSZ and authenticated using DMSZ STR profiling protocols. All cell lines tested negative for Mycoplasma contamination. For co-culture experiments, the MM cell lines and HS5 were passaged

and cultured separately for 24 h in a 75 cm² flask. After 24 h of culture and once HS5 50% confluency was observed, the MM cells were manually counted and 10 million MM cells with 90% or greater viability (assessed using Tryptan Blue staining), were transposed into 20 ml of fresh RPMI-1640 medium and either added into a 75 cm² flask containing HS5 cells or cultured alone in a new flask. After 72 h of culture, total cells were collected from each condition using adherent culture protocol and Trypsin-EDTA to remove attached cells. Cells were counted and viability was assessed. CD138+ plasma cells were then purified using anti-CD138 microbeads (Miltenyi Biotech, Auburn, United States). CD138 + MM cells and HS5 cells from each condition were then aliquoted separately to perform additional experiments including RNA extraction for RNA-seq and cell preparation for ATAC-seq.

### RNA-seq

RNA was extracted from CD138 + MM cells using the RNeasy MiniKit (Qiagen, Germantown, United States). RNA quantity was evaluated using the Qubit RNA Assay Kit (Life Technologies, Carlsbad, United States) and RNA quality was determined on the Bioanalyzer using the RNA Pico Kit (Agilent, Santa Clara, United States). We used at least 500 ng of total RNA for each sample. Next, library preparation was done with NEBNext Ultra RNA Library Prep Kit for Illumina (New England BioLabs, Ipswich, United States), was converted into a DNA library following the manufacturer's protocol. Library quantity was determined using the Qubit High Sensitivity DNA Kit and library size was determined using the Bioanalyzer High Sensitivity Chip Kit (Agilent). Finally, libraries were put through quantitative PCR using the Universal Library Quantification Kit for Illumina (Kapa Biosystems, Wilmington, United States) and run on the 7900HT Fast quantitative PCR machine (ABI, Grand Island, NY). Libraries passing quality control were diluted to 2 nM using sterile water, and then sequenced on the HiSeq 2000 system (Illumina, San Diego, United States) at a final concentration of 12 pM sequenced with paired end 75 bp reads, following the manufacturer's protocols. After data quality control, reads were aligned to the GRCh38 reference genome using STAR and transcript counts calculated using featureCounts[40,41]. Pre- and post-alignment data quality control was performed using FastQC (https://www.bioinformatics.babraham.ac.uk/projects/fastqc/). Differential gene expression analysis was performed using DESeq2 using default parameters (summary estimates across the biological replicates are reported), sample and gene clustering using heatmap.2 using default parameters with Euclidean distance (https://CRAN.R-project.org/package=gplots), and visualization of differentially expressed genes using Stata[42]. Functional annotation of the differentially expressed genes was performed using gProfiler[43].

### ATAC-seq

For each condition, 50,000 CD138[+] MM cells were aliquoted and prepared in duplicates as previously reported[28]. Cells were lysed for 10 min at 4 °C in lysis buffer (10 mM Tris-HCl pH 7.4, 10 mM NaCl, 3 mM MgCl2, 0.1% IGEPAL CA-360). After lysis, the pellets were subject to a transposition reaction (at 37 °C for 60 min) using the TD buffer and transposase enzyme (Illumina Nextera DNA preparation kit, FC-121–1030). The transposition mixture was purified using a MinElute PCR purification kit (Qiagen). Library amplification was performed using custom Nextera primers and the number of total cycles determined by running a SYBR-dye based qPCR reaction. Amplified libraries were purified using a PCR purification kit (Qiagen) and sequenced with paired end 75 bp reads on an NextSeq instrument (Illumina). After data quality control, reads were aligned to GRCh38 using bowtie2[44]. Pre- and post-alignment data quality control was performed using FastQC (https://www.bioinformatics.babraham.ac.uk/projects/fastqc/).

Aligned reads from were sorted and indexed using samtools and peaks were called using MACS2 using default parameters[44,45]. First, differentially accessible candidate genomic regions were identified using

MACS2 by treating the samples within each experimental condition as biological replicates. Second, differentially accessible genomic regions were selected from the candidate regions by implementing a DiffBind-like differential accessibility analysis by treating the peaks as individual features and the samples within each experimental condition as biological replicates (summary estimates across the biological replicates are reported)[46]. To annotate and associate these regions with potential target genes the Genomic Regions Enrichment of Annotations Tool (GREAT) was used[47]. The cis-interactions predicted by GREAT were then further filtered to avoid violation of the conserved MM topologically associating domain boundaries derived from Hi-C experiments in RPMI and U266[48]. Candidate cis-regulatory elements were validated using the ENCODE, GeneHancer, and Hacer databases and visualized using Intervene[49–52]. Functional annotation of the predicted target genes was performed using gProfiler[43]. Candidate transcription factors potentially binding these accessible regions were identified by transcription factor footprinting with TOBIAS[53]. Additional motif enrichment analyses were performed with HOMER[54].

## scRNA-seq + scATAC-seq
Co-cultured MM1S and HS5 cells were processed for single-cell Multiome ATAC + Gene Expression (10X Genomics) with a targeted nuclei recovery of approximately 10,000. Nuclei were isolated according to the Demonstrated Protocol: Nuclei Isolation for Single Cell Multiome ATAC + Gene Expression (10X Genomics, CG000365 Rev A). Approximately one million cells were added to a 5.0 mL low binding tube and centrifuged ($300 \times g$ for 5 min at 4 °C) using a swinging bucket rotor. Cells were washed twice with PBS + 0 .04% BSA and were passed through 40uM Flowmi cell strainer to remove any clumps. After pelleting the strained cells, the cell pellet was resuspended in 100 μL of chilled 10X Genomics Lysis Buffer (10 mM Tris-HCl pH 7.4, 10 mM NaCl, 3 mM MgCl2, 0.1% Tween-20, 0.1 % NP-40 Substitute, 0.01% digitonin, 1% BSA, 1 mM DTT, 1 U/μL RNase inhibitor 40 U/mL) by pipette-mixing 10 times. Cells were incubated on ice for 3 min, followed by dilution with 1 mL of chilled Wash Buffer (10 mM Tris-HCl pH 7.4, 10 mM NaCl, 3 mM MgCl2, 0.1% Tween-20, 1% BSA, 1 mM DTT, 1 U/mL RNase inhibitor 40 U/mL). Nuclei were then centrifuged ($500 \times g$ for 3 min at 4 °C), and the supernatant was slowly removed. The nuclei were washed one additional time with 1 mL Wash Buffer. Nuclei were resuspended in chilled diluted nuclei buffer (1X Nuclei Buffer, 1 mM DTT, 1 U/mL RNase inhibitor 40 U/mL); the concentration was determined using a hemocytometer and the samples were adjusted to a concentration appropriate for our targeted nuclei recovery. The single-cell ATAC library construction and gene expression library construction was carried out as described in the Chromium Next GEM Single Cell Multiome ATAC + Gene Expression User Guide (CG000338 Rev A). ATAC and GEX libraries were sequenced separately on an HiSeq 4000 (Illumina) before demultiplexing, alignment to the reference genome, and post-alignment quality control. A total of 8,293 single cells passed standard scRNA-seq and scATAC-seq quality control metrics (number of detected features, mitochondrial gene expression, transcription start site enrichment). MM1S and HS5 cells were computationally separated based on their single-cell gene expression profiles using a 1000-gene classifier. The classifier consisted of the top 500 differentially expressed genes between pure MM1S and pure HS5 cells (as measured by bulk RNA-seq). Among the MM1S cells, single-cell chromatin accessibility was analyzed and cis-regulatory interactions were identified using Cicero with default parameters[55].

## Patient populations
The 559 patients with newly diagnosed MM treated on the TT2 / TT3 clinical trial protocols with thalidomide and bortezomib containing regimens between 1998 and 2006 were used as for discovery (derivation cohort, GSE24080)[56,57]. Gene expression in the derivation cohort was measured on an Affymetrix Human Genome U133 Plus 2.0

microarray. Two hundred and fourteen patients with newly diagnosed MM treated on the IFM 2009 clinical trial protocol with a lenalidomide and bortezomib containing regimen between 2010 and 2012 were used for validation (validation cohort I). Gene expression in the validation I cohort was measured by RNA-seq. Additionally, 635 patients treated with various novel agent containing regimens from 2011 on as part of the MMRF CoMMpass study were used for validation (validation cohort II, IA16)[58]. Gene expression in the validation II cohort was measured by RNA-seq. The IA16 data was accessed through the MMRF Researcher Gateway (https://research.themmrf.org). The gene expression of 2688 circulating plasma cells and 2688 bone marrow plasma cells of patients with MM was estimated using the data reported by *Ledergor* et al. (GSE117156)[59]. The 2688 bone marrow plasma cells were randomly sampled from the 12672 bone marrow plasma cells in the data repository to balance the sample sizes for comparison. In GSE117156, single-cell gene expression was measured using MARS-seq and we used log-normalized counts as input for Gene Set Enrichment Analysis (GSEA)[60]. The gene expression of 303 extra-medullary plasma cells and 177 bone marrow plasma cells of patients with MM was estimated using the data reported by *Ryu* et al. (GSE106218)[61]. In GSE110499, single-cell gene expression was measured using the Fluidigm C1 platform and the normalized measures of single-cell gene expression (TPM) available in the data repository were used as input for GSEA.

## Σ ASI as a summary measure of adverse stromal interactions
We identified 10 stromal interaction genes independently associated with progression-free survival in the derivation cohort (GSE24080). These 10 genes were selected among the 68 genes with concordant gene expression and chromatin accessibility changes. The 10 genes represent the consensus of forward (p = 0.200 for addition to the model) and backward feature selection ($p = 0.200$ for removal from the model) in proportional hazards regression models. Five of the 10 genes (*FSTL1*, *FSCN1*, *VCAN*, *GADD45A*, and *AKAP12*) were associated with increased overall survival (HR below 1.0 in the final multivariable-adjusted proportional hazards regression model including all 10 genes). The other five genes (*AIM2*, *ZEB2*, *IL6*, *ARAP3*, and *PTK2*) were associated with decreased overall survival (HR above 1.0) in the same model. First, we quantile-normalized the expression of each of the 10 genes by creating deciles of their expression (1 = lowest expression, 10 = highest expression). The Σ ASI is defined as (the sum of deciles of expression of the adverse genes) plus (the sum of 10 minus the deciles of expression of the favorable genes). Therefore, a Σ ASI of 0 indicates the maximum expression of all favorable genes and minimum expression of all adverse genes. Conversely, a Σ ASI of 90 indicates the maximum expression of all adverse genes and minimum expression of all favorable genes. R and Stata code to calculate the Σ ASI classifier and pertinent survival analyses are provided as supplementary software (Supplementary Software 1). We dichotomized the Σ ASI at 50 designating a group without (ASI-, Σ ASI < 50) and a group with ASI (ASI + , Σ ASI ≥ 50). This dichotomization was chosen to achieve approximately a 70/30 split between standard-risk and high-risk patients. Consequently, approximately one third of the patients was classified as ASI+ in all cohorts (derivation = 29%, validation I = 32%, validation II = 30%).

## Data analysis
Data are presented as median (range) unless denoted otherwise. Medians were the preferred measure of central tendency and non-parametric hypothesis tests were used for comparisons unless stated otherwise. Continuous variables were compared using the Mann-Whitney-U test (difference between two groups), the Kruskal-Wallis test (any difference between more than two groups), or Cuzick's non-parametric test for trend (trend across more than two groups)[62–64]. Categorical variables were compared using Fisher's exact test[65]. The statistical significance of regression coefficients was evaluated using

the Wald test[66]. All hypothesis tests (other than the hypergeometric tests) were two-sided and *p*-values below 0.05 were considered statistically significant. The method described by *Benjamini and Hochberg* was used to control the false discovery rate (FDR)[67]. Overall and progression-free survival estimates were calculated using the method described by *Kaplan and Meier*[68]. Overall survival was defined as the time from diagnosis to death and patients who were alive at the end of follow-up were censored. Progression-free survival was defined as the time from diagnosis to disease progression or death. Patients who were alive and had not progressed at the end of follow-up were censored. The log-rank test was used to compare time to event data across subgroups[69]. Multivariable-adjusted (Cox) proportional hazards regression models were used to assess the association between the covariates of interest and survival outcomes[70]. Violations of the proportional hazards assumption were evaluated using scaled Schoenfeld residuals[71]. Logistic regression was used to assess the association between covariates and response to therapy. Early therapeutic resistance was defined as Stable Disease (SD) or Progressive Disease (PD) after three months of treatment[72]. The EMC-92 and UAMS-70 high-risk classifiers were calculated as described by *Kuiper* et al. and *Shaughnessy* et al., respectively[37,73]. The overlap between gene sets was visualized using venn (https://CRAN.R-project.org/package=venn). Circular scatter plots were generated using circlize[74]. R was used for data processing and analysis, Stata for visualization[42,75].

## Statistics & reproducibility

The sample size was chosen based on the RNA-seq power calculation (at least 80% power to detect a 3-fold or greater change in gene expression at an α-level of 5%). Average sequencing coverage and the coefficient of variation were empirically derived from a large number of human RNA-seq experiments[76]. No data were excluded from the analyses. The experiments were not randomized. The Investigators were not blinded to allocation during experiments and outcome assessment.

## Reporting summary

Further information on research design is available in the Nature Portfolio Reporting Summary linked to this article.

## Data availability

The raw bulk and single-cell RNA and ATAC sequencing data generated in this study have been deposited in the Gene Expression Omnibus (GEO) database under accession code GSE220144. Publicly available datasets analyzed during the current study are available in Gene Expression Omnibus: GSE2658 (gene expression microarray data)[37] [https://www.ncbi.nlm.nih.gov/geo/query/acc.cgi?acc=GSE2658], GSE106218 (single-cell gene expression data)[61] [https://www.ncbi.nlm.nih.gov/geo/query/acc.cgi?acc=GSE106218], GSE24080 (gene expression microarray data)[77] [https://www.ncbi.nlm.nih.gov/geo/query/acc.cgi?acc=GSE24080], GSE117156 (single-cell gene expression data)[59] [https://www.ncbi.nlm.nih.gov/geo/query/acc.cgi?acc=GSE117156], and GSE110499 (single-cell gene expression data)[61] [https://www.ncbi.nlm.nih.gov/geo/query/acc.cgi?acc=GSE110499]. The MMRF IA16 bulk gene expression data was accessed through the MMRF Researcher Gateway (https://research.themmrf.org)[58]. The IFM dataset analyzed during the current study are available from the authors upon reasonable request. The remaining data are available within the Article file, Supplementary Information file, Supplementary Data files, or Source Data file. Source data are provided with this paper.

## Code availability

R and Stata code to calculate the Σ ASI classifier and pertinent survival analyses are provided as Supplementary Software (**Supplementary Software 1**).

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

## Acknowledgements

This work was supported by NIH grants P01-155258 and 5P50 CA100707 (N.C.M. and M.K.S.); Department of Veterans Affairs Merit Review Award I01BX001584-01 (N.C.M.), and a Riney Family Foundation grant.

## Author contributions

M.B., M.K.S. and N.C.M. designed the study. R.E.S., S.T. and M.F. performed experiments. H.A.L. and N.C.M. contributed clinical data and patient samples. M.B., G.P., M.K.S. and N.C.M. analyzed the data. M.B., G.P., M.K.S. and N.C.M. wrote the manuscript. All authors critically revised and approved the final version of the manuscript.

## Competing interests

N.C.M. is a consultant for BMS, Janssen, Oncopep, Amgen, Pfizer, Karyopharm, Legend, NextRNA, Raqia, Abbvie, Takeda, and GSK, and on the board of directors of Oncopep. G.P. is a co-founder of Phaeno Biotechnologies, serves on the Scientific Advisory Board of Konica Minolta Healthcare Americas and consults with Foundation Medicine and Delfi Diagnostics. The remaining authors declare no competing interests.
