## [Peer Review File · Nature Communications]

Bone Marrow Stromal Cells Induce Chromatin Remodeling in Multiple Myeloma Cells Leading to Transcriptional ChangesReviewers' Comments:

Reviewer #1:

Remarks to the Author:

Comments to authors

In this manuscript, Binder et al. first show that multiple myeloma (MM) cell lines after interaction with a bone marrow stromal cell line, experience chromatin remodeling of cis-regulatory elements which are associated with increased transcription of their target genes, mainly involved in cell migration and cytokine signaling. Interestingly, the transcriptional profile of those genes in myeloma cells after stromal interaction was found to be characteristic of MM extramedullary disease, and also defined a high-risk group of myeloma patients associated with accelerated disease dissemination, early development of therapy resistance, and poor survival outcomes. Moreover, the authors select a subset of 10 of those stromal interaction genes with independent prognostic significance on survival outcomes and response to treatment in newly diagnosed MM.

The results provided in the manuscript hold significance not only for the knowledge of myeloma disease but also because of the relevance of their clinical implications. By integrating gene expression and chromatin accessibility in their in vitro studies, authors mechanistically link the upregulated expression of at least some genes in myeloma cells after stromal interaction to increased chromatin accessibility of enhancers of those genes. Besides, this transcriptional program of myeloma cells after stromal interaction offers the possibility of identifying a subtype of high-risk MM patients with extramedullary disease characteristics, for whom identification of therapeutic targets would be of great importance due to their poor prognosis. Prediction of resistance to VRd (bortezomib, lenalidomide, and dexamethasone) treatment in three different patient cohorts was also possible based on the expression of the ASI (adverse stromal interaction) genes.

The manuscript is based on appropriate RNA-seq and ATAC-seq technology in the in vitro studies, followed by integration and analysis of data to associate differentially accessible regions with their target genes. Subsequent analyses of those "stromal interaction genes" are then used to select a subgroup of 10 genes based on their independent prognosis significance. The expression of those genes was strongly associated with survival outcomes and early therapeutic resistance to VRd treatment in three separate patient datasets from newly diagnosed MM patients, which underscores the validity and robustness of the performed analyses in the manuscript. In general, enough context is provided to understand the presented results in the Introduction section, and based on their results authors come to interesting deductions for patients with extramedullary disease phenotype at the moment of diagnosis. The presented results well support the conclusions in the manuscript.

Nevertheless, there are some issues that may be considered by the authors:

a) In their initial in vitro co-culture setup, three different myeloma cell lines (MM1S, RPMI8226 and INA-6) are co-cultured with the bone marrow (BM) stromal cell line HS-5. Have authors found transcriptional or chromatin accessibility differences after stromal interaction between the three cell lines? This information should be included in the manuscript ("Material and methods section" and "Results").

b) Although data from the manuscript clearly define the stromal-induced interaction genes as recapitulating the transcriptional program of extramedullary disease, this might be somewhat expectable considering that myeloma cell lines are established from patients at plasma cell leukemia stages, and thus are "per se" representative of extramedullary disease. This issue may be at least commented on by the authors in the Discussion section.

c) Similarly, co-cultures are established with the HS-5 cell line, which is a normal donor BM-derived stromal cell line. Would interaction with an MM-derived BM stromal cell have rendered the same transcriptional changes in myeloma cell lines after interaction? This may also be discussed by the authors.

d) In the manuscript, the presence of adverse stromal interactions (ASI+) was found predictive of

therapy resistance for patients being treated with VRd regimens. It would also be of interest to check for the predictive power of ASI+ for resistance to newer targeted therapies and/or immunotherapies in other patient cohorts.

e) Although references from the same group are included in the "Introduction" section (e.g. ref 17), authors may cite McMillin et al 2013, Nat Rev Drug Discov 12: 21728, which is a comprehensive study of deregulation of gene expression in myeloma cells after interaction with stromal cells.

g) BM mesenchymal stromal cells have also been reported to modify their transcriptional profile after interaction with myeloma cells [e.g. Garcia-Gomez et al 2014, Oncotarget 5:8284-305], and for certain genes, stromal transcriptional changes are also mediated through chromatin modifications in direct relation to the suppression of osteogenic genes [e.g. Adamik et al 2019, JBMR 3:e10183; Adamik et al 2018, J Bone Oncol 13: 62-79]. This issue may also be commented by authors if found pertinent.

h) Please review the Abstract (eg. line 28) to clearly reflect that chromatin remodeling and transcriptional changes induced by stromal interaction, correspond to myeloma cell lines. Similarly, Figure 1d should be "Stroma-induced CREs" and not "Stroma CREs".

h) Minor issues: please review the "Competing interests" section for the identity of acronyms.

Typos or grammatical errors: page 4, lines 97-98; page 12, line 362; page 24, line 669; page 24, lines 679-680.

Reviewer #2:

Remarks to the Author:

In the manuscript by Binder et al entitled 'Bone Marrow Stromal Cell Induced Chromatin Remodeling and Associated Transcriptional Changes in Multiple Myeloma', the authors set to address the role that stroma cells have into multiple myeloma cells. The assumption was that that interaction could induce changes that allowed MM cells to progress/ migrate, and be ultimately responsible for inducing BM niche independency and as consequence the extra-medullary characteristics.

In order to address it, they co-cultured MM cell lines (MM1.S, RPMI-8226 and INA-6) with and without a BM stroma cell line (HS5) for 72hrs. After this time, they evaluated the changes in the transcriptome by RNA-seq and chromatin accessibility (ATAC-seq).

They defined differentially accessible chromatin areas (DA 4511) and differentially expressed genes (DE-224 genes) and did gene set enrichment analysis to define the pathways/functions that were more likely to be affected by the co-culture.

With the DA ones they define potential target genes (independent of DE-4626 genes for 4288 regulatory regions). By motif analysis, they predicted transcription factors that potentially bind/regulate those regions.

Subsequently integrated both data sets to define the potential regulatory regions responsible for the changes of expression. They define 68 genes that were DE and have a differentially accessible region associated.

From those 68 genes they identify 10 genes that when differentially expressed (up or down) were associated with survival and define a stroma signature (ASI) that then went to test against different data set to establish its prognostic significance (survival, response to treatment, extra medullar disease) independent of other factors.

It is an interesting manuscript and the findings can also be considered interesting but not novel. There are several major/minor concerning point:

1. The idea of stroma influencing the behaviour (and in this case chromatin accessibility) of MM cells, is not a new concept/idea and has been already tested in an article/s that the authors fail to reference. Dziadowicz et al. 'Bone Marrow Stroma-Induced Transcriptome and Regulome Signatures of Multiple Myeloma'. Cancers PMID: 35205675 04 march 2021. In this article they use the same (and more complex system) using 2/3 identical cell lines. The authors should at least acknowledge the paper and validate their data with this previous article (RNA-seq and ATAC-seq). There is an older

article that could also been used to validate the DE genes (using microarrays) by McMillin et al (Coauthored by one of the corresponding authors)- PMID: 20228816. They also failed to acknowledge and compare their data

2. There is an over-interpretation of the data:

- The data has been acquired in an in vitro model with MM cell lines that (as all current cell lines) have been adapted to grow independently of stroma. So they may already represent an advanced/EMD stage (independent of the microenvironment signals)
- The Stroma used is also a cell line.

Both conditions do not necessarily negate the findings but the caveats of the in vitro system should be acknowledged and discussed. In several parts of the paper there is no mention that it is an in vitro system.

3. There is no validation of any of the molecular findings. The authors defined CRE-gene target by prediction analysis, but fail to do any type of validation: reporter assay, KO/ downregulation (CRISPR mediated), 4C analysis of the open regions to show that in fact there is a gained accessibility that is responsible for the activation of certain genes (any of the 5 upregulated genes?). After all, the authors are exploring the changes in chromatin accessibility and the associated transcriptional change (as for the title). The findings do not support such a title

4. There is no validation of the possible function of any of the 10 genes important for prognosis. This could be very interesting. Showing that forced upregulation of the genes confers migratory potential to the cells (in vitro or in animal models), could support their findings.

5. The authors used 3 different MM cell lines, what is the rationale behind choosing those in particular? Different genetic backgrounds? They failed to show the correlation of the data between the 3 cell line models. Do all 3 behave the same? Are the DE genes shared to all 3? Similarly are the chromatin accessibility changes similar? This needs at least clarification.

If their effect was independent of the genetic alteration s/genetic background (as they claimed), more/better analysis should have been done.

6. The regulome of primary MM, B cell development and cell lines has been defined, not only super enhancers should be used to compare the differentially accessible regions and their potential targets. Even by published chromatin conformation analysis.

7. The potential TF regulating the putative enhancers have been defined by motif analysis. ATAC-seq allows to study the foot-printing of potential TFs, and this that can be a better/validated approach. The public databases of human ChIP-seq does not give any important information, unless they are on MM cells (or closely correlated). TF bind differently in different cellular background, and more over under different stimuli, as the manuscript want to show.

Importantly (and not addressed) TF have a very selective expression pattern. SO the use of general ChIP-seq databases is not a good reference.

8. It is not clear why in figure 2 g-h you show all the differentially accessible CRE but correlate to all the putative genes regardless if they are DE expressed (as you have only 224 not the thousands that are reported in the figure).

Please clarify what genes are being considered for the analysis as the take home message is very different. What is the added value of the analysis if not just to show that they are prototypical CRE? They could be part of Supplementary information?

Probably the most interesting data is the identification of the stromal signature with potential prognosis capabilities. Here the authors failed to validate the ASI signature with other stroma data sets (papers cited above) and with RNA-seq/ATAC data sets from primary sources

Minor:

The TFs mentioned as in lane 108 are different from the ones discussed in lane 224. IRF4 and PBX3 are not discussed anywhere in the article. Are they predicted to bind any of the CRE regulating ASI?

Lane 104 the gene ontology analysis shows several functions and pathways that are interesting but general not as part of specific MM datasets. The sentence is misleading.

Fig 2C and 2J lacks coordinates

Number of genes stated as 4626 (lane 104) or 4646 (lane 690)

Lane 61 and 65, lack of reference

Why is 72 hrs of co-culture used. DO the MM cells acquire functional changes migration, adhesion ?)

Response to Reviewer #1 (expertise in bone marrow stromal cells and MM)

- 1. In their initial in vitro co-culture setup, three different myeloma cell lines (MM1S, RPMI8226 and INA-6) are co-cultured with the bone marrow (BM) stromal cell line HS-5. Have authors found transcriptional or chromatin accessibility differences after stromal interaction between the three cell lines? This information should be included in the manuscript (“Material and methods section” and “Results”).**

Indeed, there are differences between the three cell lines in terms of transcriptional and accessibility changes in response to stromal co-culture. In general, the changes in accessibility are more heterogenous than the changes in gene expression. We first examined the gene expression profiles of the three cell lines before and after stromal co-culture using pairwise correlation (*new Supplementary Figure 1b-d, revised manuscript lines 120-127 and 843-857*). Gene expression before and after stromal co-culture are highly correlated ($r > 0.90$ and $p < 0.001$ for all three cell lines). Next, we categorized transcripts as down-regulated (decrease in normalized transcript count), stable (no change in normalized transcript count), and up-regulated (increase in normalized transcript count) after stromal co-culture for each cell line. There was strong agreement between the three cell lines in terms of the transcriptional changes in response to stromal co-culture (inter-rater agreement > 0.70 and $p < 0.001$ for all three cell lines). We defined the transcripts with discordant behavior in response to stromal co-culture ($n = 4083$ transcripts, 6.73% of all transcripts) as those outside Bland-Altman limits of agreement and provide them as supplementary data along with relevant functional annotation (*revised Supplementary Table 1*). The changes in accessibility are more heterogenous between the three cell lines (*new Supplementary Figure 1a*) showing three times more newly accessible genomic regions after stromal co-culture in MM1S (compared to RPMI) and four times more in RPMI (compared to INA6). Potential sources of this observed heterogeneity are the different genetic backgrounds of the three cell lines. The localization (middle panel) and frequency by chromosome of these regions uniquely accessible in the three cell lines are shown in **Supplementary Figure 1a**. We provide these regions along with relevant functional annotation as supplementary data (*revised Supplementary Table 1*). **Supplementary Figure 1 has been revised to include these data, Supplementary Table 1 has been revised, and the “Results” section has been amended accordingly (*revised manuscript lines 120-127 and 843-857*).**

- 2. Although data from the manuscript clearly define the stromal-induced interaction genes as recapitulating the transcriptional program of extramedullary disease, this might be somewhat expectable considering that myeloma cell lines are established from patients at plasma cell leukemia stages, and thus are “per se” representative of extramedullary disease. This issue may be at least commented on by the authors in the Discussion section.**

This is a very important point. To better understand to which extent myeloma cell lines are characterized by stromal gene expression to begin with, we investigated the expression of the 68 stromal genes across 70 commonly used myeloma cell lines at baseline (without stromal co-culture). Only two of the 70 myeloma cell lines (3% of all cell lines, namely CAG and SUHRR) showed increased expression in $> 50\%$ of the stromal genes (*new Supplementary Figure 1e, revised manuscript lines 138-140*). Likewise, we did not observe an enrichment in the expression of the 68 stromal genes in myeloma cell lines with high-risk IGH translocations (Enrichment Score -0.57 , $p = 0.644$, *new Supplementary Figure 1f, revised manuscript lines 138-140*). From these data, we conclude that stromal gene expression is not a defining feature of myeloma cell lines (i.e. not *per se* representative of extramedullary disease). Even though all of the myeloma cell lines were established from patients in advanced disease stages (highly proliferative, leukemic, potentially disseminated with extramedullary manifestations), 97% of them did not show increased gene expression of the genes in question at baseline. **These data have been added as a supplementary figure (*revised manuscript lines 138-140 and 843-857*).** The “Results” and “Discussion” sections have been amended accordingly, as requested (*revised manuscript lines 315-319*).

- 3. Similarly, co-cultures are established with the HS-5 cell line, which is a normal donor BM-derived stromal cell line. Would interaction with an MM-derived BM stromal cell have rendered the same transcriptional changes in myeloma cell lines after interaction? This may also be discussed by the authors.**

In these experiments, we have used the stromal cell line to have uniform interaction and have not attempted to isolate stromal cells from patient samples to perform co-culture experiments. However, we agree with this point and have modified our discussion to reflect this, as requested (*revised manuscript lines 315-319*).

- 4. In the manuscript, the presence of adverse stromal interactions (ASI+) was found predictive of therapy resistance for patients being treated with VRd regimen. It would also be of interest to check for the predictive power of ASI+ for resistance to newer targeted therapies and / or immunotherapies in other patient cohorts.**

We agree that the effect of stromal gene expression on response to CAR T-cell therapies and bispecific antibody therapies is of great interest. However, currently there is paucity of publicly available data to address this question. In the manuscript, we reference previous studies demonstrating the ability of stromal cells to protect myeloma cells from immune-related injury (suggesting a potential role in resistance to cellular therapies). The “Discussion” section has been extended to include this, as requested (*revised manuscript lines 380-383*).

- 5. Although references from the same group are included in the “Introduction” section (e.g. ref 17), authors may cite McMillin et al 2013, Nat Rev Drug Discov 12: 21728, which is a comprehensive study of deregulation of gene expression in myeloma cells after interaction with stromal cells.**

The “Introduction” and “Discussion” sections have been amended to include this reference, as requested (*revised manuscript lines 80-82, 361-364, and 378-380*).

- 6. BM mesenchymal stromal cells have also been reported to modify their transcriptional profile after interaction with myeloma cells [e.g. Garcia-Gomez et al 2014, Oncotarget 5:8284-305], and for certain genes, stromal transcriptional changes are also mediated through chromatin modifications in direct relation to the suppression of osteogenic genes [e.g. Adamik et al 2019, JBMR 3:e10183; Adamik et al 2018, J Bone Oncol 13: 62-79]. This issue may also be commented by authors if found pertinent.**

The “Introduction” section has been amended to include these references (*revised manuscript lines 82-90*).

- 7. Please review the Abstract (eg. line 28) to clearly reflect that chromatin remodeling and transcriptional changes induced by stromal interaction, correspond to myeloma cell lines. Similarly, Figure 1d should be “Stroma-induced CREs” and not “Stroma CREs”.**

This has been corrected in the “Abstract” section and Figure 1d, as requested (*revised manuscript lines 31 and 893*).

- 8. Minor issues: please review the “Competing interests” section for the identity of acronyms. Typos or grammatical errors: page 4, lines 97-98; page 12, line 362; page 24, line 669; page 24, lines 679-680.**

Acronyms have been corrected (*revised manuscript lines 719-725*). The manuscript has been revised for grammar and orthography.

Response to Reviewer #3 (expertise in MM genomics and epigenomics)

1. The idea of stroma influencing the behaviour (and in this case chromatin accessibility) of MM cells, is not a new concept / idea and has been already tested in an article/s that the authors fail to reference. Dziadowicz et al. 'Bone Marrow Stroma-Induced Transcriptome and Regulome Signatures of Multiple Myeloma'. Cancers PMID: 35205675 04 march 2021. In this article they use the same (and more complex system) using 2/3 identical cell lines. The authors should at least acknowledge the paper and validate their data with this previous article (RNA-seq and ATAC-seq). There is an older article that could also been used to validate the DE genes (using microarrays) by McMillin et al (Coauthored by one of the corresponding authors)- PMID: 20228816. They also failed to acknowledge and compare their date.

We appreciate the comment and have now corrected our oversight by including both articles as reference and in discussion (*revised manuscript lines 80-90 and 304-307*).

2. There is an over-interpretation of the data:

- The data has been acquired in an in vitro model with MM cell lines that (as all current cell lines) have been adapted to grow independently of stroma. So they may already represent an advance/EMD stage (independent of the microenvironment signals)

- The Stroma used is also a cell line. Both conditions do not necessarily negate the findings, but the caveats of the in vitro system should be acknowledged and discussed. In several parts of the paper there is no mention that it is an in vitro system.

This is an important point. We have now revised our discussion to avoid over-interpretation of the presented results and to discuss the question about role of cell line in these experiments as explained above in response to comment #2, by reviewer 1.

3. There is not validation of any of the molecular findings. The authors defined CRE-gene target by prediction analysis, but fail to do any type of validation: reporter assay, KO / downregulation (CRISPR mediated), 4C analysis of the open regions to show that in fact there is a gained accessibility that is responsible for the activation of certain genes (any of the 5 upregulated genes?). After all, the authors are exploring the changes in chromatin accessibility and the associated transcriptional change (as for the title). The findings do not support such a title.

As suggested by the reviewer, we performed a single-cell multi-omics (combined profiling of RNA seq and ATAC seq from the same cells) to correlate our findings from bulk genomics to large number of single cells. Using these new data, we demonstrate that all 10 predicted enhancer-target gene pairs of the 10-gene signature have positive single-cell co-accessibility (*new Figure 3c, new Supplementary Figure 2d*). The "Results" and "Methods" sections were revised to reflect these additional supporting data (*revised manuscript lines 208-212, 577-811, 962-966, and 1060-1074*).

4. There is no validation of the possible function of any of the 10 genes important for prognosis. This could be very interesting. Showing that forced upregulation of the genes conferees migratory potential to the cells (in vitro or in animal models), could support their findings.

As with other reports describing prognostic markers in myeloma, such as 70-gene or 92-gene signatures, we here report 10 genes whose expression is modulated by MM-stromal interactions and are predictive of outcome and extramedullary disease indicative of potential biological role. We agree that studying possible functions of these genes will be potentially interesting. Function of 4 of these genes have been already studied by us or others (IL-6, PTK2, AIM2, and GADD45A) and two additional genes have been described previously to have role in myeloma biology / prognosis (AKAP12 and VCAN). We are now including this information in the discussion (*revised manuscript lines 361-367*). We believe further validation of these genes requires in-depth molecular and

cell signaling investigation and hope that this publication will provide the required initial basis for their further investigation.

5. The authors used 3 different MM cell lines, what is the rationale behind choosing those in particular? Different genetic backgrounds?

The reviewer's supposition is spot on. The myeloma cell lines were chosen for their representative genetic background. The genetic features considered were IGH translocations (MM.1S); p53 mutation (RPMI) and loss of p53 and t(11:14) (INA-6). This is now added in the methods as the rationale for selection of the cell lines (*revised manuscript lines 495-496*).

6. They failed to show the correlation of the data between the 3 cell line models. Do all 3 behave the same? Are the DE genes shared to all 3? Similarly, are the chromatin accessibility changes similar? This needs at least clarification. If the effect was independent of the genetic alterations / genetic background (as they claimed), more / better analysis should have been done.

As described above in Comment 1, Rev. 1, additional analyses have been performed and supplementary data have been added, as requested. (*new Supplementary Figure 1b-d, revised manuscript lines 120-127 and 843-857*)

7. The regulome of primary MM, B cell development and cell lines has been defined, not only super enhancers should be used to compare the differentially accessible regions and their potential targets. Even by published chromatin conformation analysis.

This was not quite clear in the original manuscript. The presented analysis includes all differentially accessible regions, not just those considered super enhancers. The analysis originally presented in **Figure 1e** examined the subset of super-enhancers among all identified enhancers. *This subgroup analysis (super-enhancers only) has been removed and the manuscript text has been revised for clarity (revised manuscript lines 122-123, 893, and 907).*

8. The potential TF regulating the putative enhancers have been defined by motif analysis. ATAC-seq allows to study the foot-printing of potential TFs, and this that can be a better/validated approach.

We have changed the analysis to select the candidate transcription factors by transcription factors footprinting rather than motif analysis, as requested (*revised manuscript lines 130-132 and 911-914*). Motif analysis results are shown in addition to the footprinting results as both approaches agree on several of the top candidates.

9. The public databases of human ChIP-seq does not give any important information, unless they are on MM cells (or closely correlated). TF bind differently in different cellular background, and more over under different stimuli, as the manuscript want to show. Importantly (and not addressed) TF have a very selective expression pattern. So the use of general ChIP-seq databases is not a good reference.

We have **removed** this analysis due to its lack of plasma cell specificity (*revised manuscript lines 893*).

10. It is not clear why in figure 2 g-h you show all the differentially accessible CRE but correlate to all the putative genes regardless if they are DE expressed (as you have only 224 not the thousands that are reported in the figure). Please clarify what genes are being considered for the analysis as the take home message is very different. What is the added value of the analysis if not just to show that they are prototypical CRE? They could be part of Supplementary information?

The purpose of this analysis is indeed to show that these are prototypical CRE and *these results have been moved to Supplementary Figure 2a-c, as suggested (revised manuscript lines 1061-1092)*. The applied FDR

cut-off of 0.05 is common, but arbitrary. For the purpose of demonstrating prototypical CRE behavior, genes above the FDR cut-off have been included in this analysis.

11. Probably the most interesting data is the identification of the stromal signature with potential prognosis capabilities. Here the authors failed to validate the ASI signature with other stroma data sets (papers cited above) and with RNA-seq / ATAC data sets from primary sources.

We highlight independent prognostic significance of the identified genes in the 3 major publicly available gene expression data sets of newly diagnosed multiple myeloma patients treated with contemporary therapeutic regimens: MMRF (n=635), UAMS (n=559), and IFM (n=214). Our independent validation includes more than 1400 newly diagnosed MM patients, all receiving combination therapies with both immunomodulators and proteasome inhibitors. Importantly, we show the independent prognostic significance of our findings when adjusting for stage, high-risk cytogenetics, and other high-risk gene expression classifiers (EMC-92, UAMS-70). Importantly, we have focused on prognosis in newly-diagnosed patients. However, currently no data set with RNA-seq / ATAC from primary sources is available for such analysis. McMillin et al. paper which is from our group has looked at expression changes in MM with stromal interaction, but in that paper we had focused on prognostic significance in patients with relapsed multiple myeloma treated with single-agent high-dose dexamethasone alone, but not in patients treated with single-agent bortezomib (the other arm of the APEX trial). Dziadowicz et al. identified a 232-gene signature but did not investigate the prognostic significance of their 232-gene signature. We have revised the manuscript to discuss these results (*revised manuscript lines 80-83 and 304-307*).

12. Minor changes:

The TFs mentioned as in lane 108 are different from the ones discussed in lane 224. IRF4 and PBX3 are not discussed anywhere in the article. Are they predicted to bind any of the CRE regulating ASI?

The manuscript has been revised to be more consistent between the two sections (*revised manuscript lines 130-132 and 310-312*).

Lane 104 the gene ontology analysis shows several functions and pathways that are interesting but general not as part of specific MM datasets. The sentence is misleading.

The sentence has been revised to reflect the source of the annotation data (*revised manuscript lines 151-154, 194-196, 946-950, 958-962, revised Supplementary Table 3*).

Fig 2C and 2J lacks coordinates

The figure has been revised (*revised manuscript line 935*).

Number of genes stated as 4626 (lane 104) or 4646 (lane 690)

This typographical error has been corrected (*revised manuscript line 959*).

Lane 61 and 65, lack of reference

References for these two statements were added (*revised manuscript lines 73 and 77*).

Why is 72 hours of co-culture used. Do the MM cells acquire functional changes migration, adhesion?

We selected 72-hours to reflect more consistent changes not transient. This is added in the manuscript (*revised manuscript line 505*).

Reviewers' Comments:

Reviewer #1:

Remarks to the Author:

In this reviewed version "R1 NCOMMS-23-00840A", the authors have thoroughly addressed the comments and considerations that were raised in the first version of the manuscript. Concordant changes in the text and figures of the manuscript have accompanied their responses. Specifically, their reviews of point 1 has led to new Supplementary Figures 1a-f, and Supplementary Tables 1 and 2. Their reviews of point 2 has been reflected in Supplementary Figures 1e and 1f.

Nevertheless, in this revised version there are still some issues that require further consideration by the authors:

1. Please explain the difference between target genes 1 and 2 in the last bar chart in Figure 1e (page 24, lines 904-908)
2. Data from Supplementary Figures 1 e-f (page 5, lines 154-160) should be introduced later in the manuscript since they refer to the expression of the 68 stroma-induced genes in myeloma cell lines. Those 68 upregulated genes with concordant CRE accessibility are identified in the subsequent Figure 2d (page 5, line 162).
3. Please specify the color code for the high/standard risk classification in the myeloma cell lines and which high-risk cytogenetic abnormalities were considered (Supplementary Figure 1f).
4. Please check whether the actual Supplementary Figure 2d and 2e legends are interchanged.
5. It is my understanding that in this revised version, Supplementary Figures 2a-c do not correspond to the One possibility would be to maintain Supplementary Figure 2 with actual Supp Figs 2a-c, and then add a Supplementary Figure 3 including actual Supp Figs 2d-i (all related to the 10 prognostically significant stromal interaction genes).
6. As authors have explained, differences were found between the three cell lines in terms of transcriptional and accessibility changes in response to stromal co-culture (with more agreement between the three cell lines in transcriptional changes and being more heterogeneous in accessibility). Authors may also specify to which cell line the general description of the results in transcription/accessibility corresponds, or whether they are showing the common data for the three cell lines or a compendium of those.
7. Please check for correctness by referencing "Figure 4e" (page 7, line 264) instead of Figure 4c. Also in Figure 4c, check for the correctness of FDR values of scatter plot. Figure 4e lacks ordinate axis units and title.
8. Please check for assay correctness between the two sentences on page 9, line 312.
9. Minor errors: page 4, lines 116-117; page 26, line 942; page 28, line 981.

Reviewer #2:

Remarks to the Author:

Dear Authors,

After careful revision of your revised manuscript, I acknowledge that the changes introduced have made the manuscript clearer, sounder and stronger.

I still have two comments on the following points:

3. There is not validation of any of the molecular findings. The authors defined CRE-gene target by prediction analysis, but fail to do any type of validation: reporter assay, KO / downregulation (CRISPR mediated), 4C analysis of the open regions to show that in fact there is a gained accessibility that is responsible for the activation of certain genes (any of the 5 upregulated genes?). After all, the authors are exploring the changes in chromatin accessibility and the associated transcriptional change (as for the title). The findings do not support such a title.

As suggested by the reviewer, we performed a single-cell multi-omics (combined profiling of RNA seq and ATACseq from the same cells) to correlate our findings from bulk genomics to large number of

single cells. Using these new data, we demonstrate that all 10 predicted enhancer-target gene pairs of the 10-gene signature have positive single-cell co-accessibility (new Figure 3c, new Supplementary Figure 2d). The "Results" and "Methods" sections were revised to reflect these additional supporting data (revised manuscript lines 208-212, 577-811, 962-966, and 1060-1074).

The additional scRNA/ATAC -seq data clarifies the fact that certain differential accessibility regions correlate to differential expression in single cells, demonstrating that the effect is not a bulk effect and that the opening/closing of chromatin and the increased/decreased expression of the PUTATIVE target genes happens indeed in the same cell. These experiments do not validate the CRE-gene targets pairs, as it has not been demonstrated that such region is actually regulatory, or the direct link with this regulatory region and the predicted/assigned gene target. I understand that the experiments to demonstrate the functionality of the regulatory region go beyond the scope of this study, but I would call these are 'potential' CREs-gene targets.

11. Probably the most interesting data is the identification of the stromal signature with potential prognosis capabilities. Here the authors failed to validate the ASI signature with other stroma data sets (papers cited above) and with RNA-seq / ATAC data sets from primary sources. We highlight independent prognostic significance of the identified genes in the 3 major publicly available gene expression data sets of newly diagnosed multiple myeloma patients treated with contemporary therapeutic regimens: MMRF (n=635), UAMS (n=559), and IFM (n=214). Our independent validation includes more than 1400 newly diagnosed MM patients, all receiving combination therapies with both immunomodulators and proteasome inhibitors. Importantly, we show the independent prognostic significance of our findings when adjusting for stage, high-risk cytogenetics, and other high-risk gene expression classifiers (EMC-92, UAMS-70).

Importantly, we have focused on prognosis in newly-diagnosed patients. However, currently no data set with RNA-seq / ATAC from primary sources is available for such analysis. McMillin et al. paper which is from our group has looked at expression changes in MM with stromal interaction, but in that paper we had focused on prognostic significance in patients with relapsed multiple myeloma treated with single-agent high-dose dexamethasone alone, but not in patients treated with single-agent bortezomib (the other arm of the APEX trial).

Dziadowicz et al. identified a 232-gene signature but did not investigate the prognostic significance of their 232-gene signature. We have revised the manuscript to discuss these results (revised manuscript lines 80-83 and 304-307).

The authors addressed this point in a different way that I intended. What I wanted to be clarified was: Are the DE genes (68 or some of the 10 selected) found in this study also found DE in previous studies? If so, this is an independent validation of the experimental approach and I some cases with the use of different cell lines in presence of stroma.

Minor/format

I would suggest the authors to revise the colour schemes as the same colours (blue/pink-red) are used to indicate:

- +/- stroma (treatment)
- DE and DA (up/down)- gene expression
- Prognostic significance and ASI-/+

As this can be confused with the fact that + stroma also implied increase accessibility and increase gene expression, and it is clear that the treatment induce both changes (Da and DE)

Point-by-point Responses to Comments

Response to Comments by Reviewer #2

1. **Please explain the difference between target genes 1 and 2 in the last bar chart in Figure 1e (page 24, lines 904-908).**

The bars represent the frequency of cis-regulatory elements that have either one potential neighboring target gene (left bar) or two potential neighboring target genes (right bar) within their topologically associating domain (figure below adapted from the GREAT Documentation website):

We have revised the legend of **Figure 1e** to explain this in more detail (revised manuscript line 751-752).

2. **Data from Supplementary Figures 1 e-f (page 5, lines 154-160) should be introduced later in the manuscript since they refer to the expression of the 68 stroma-induced genes in myeloma cell lines. Those 68 upregulated genes with concordant CRE accessibility are identified in the subsequent Figure 2d (page 5, line 162).**

To align with the main figures and text, we have moved these supplementary data to **Supplementary Figure 2** (revised manuscript line 912-917).

3. **Please specify the color code for the high/standard risk classification in the myeloma cell lines and which high-risk cytogenetic abnormalities were considered (Supplementary Figure 1f).**

We have revised the legend of **Supplementary Figure 2b** to explain the color coding and to list the specific high-risk IGH translocation included in the group assignment (revised manuscript line 916-917).

4. **Please check whether the actual Supplementary Figure 2d and 2e legends are interchanged.**

We apologize for the oversight. The figure legends were indeed flipped. We have revised the figure legends accordingly. These supplementary data have been moved to Supplementary Figure 3a-b to align with the main figures and text (revised manuscript line 949-953).

5. **It is my understanding that in this revised version, Supplementary Figures 2a-c do not correspond to the One possibility would be to maintain Supplementary Figure 2 with actual Supp Figs 2a-c, and then add a Supplementary Figure 3 including actual Supp Figs 2d-i (all related to the 10 prognostically significant stromal interaction genes).**

To align with the main figures of the manuscript, **Supplementary Figure 2** has been split into **Supplementary Figure 2** (revised manuscript line 912-925) and **Supplementary Figure 3** (revised manuscript line 948-958).

6. **As authors have explained, differences were found between the three cell lines in terms of transcriptional and accessibility changes in response to stomal co-culture (with more agreement between the three cell lines in transcriptional changes and being more heterogeneous in accessibility). Authors may also specify to which cell line the general description of the results in transcription/accessibility corresponds, or whether they are showing the common data for the three cell lines or a compendium of those.**

Differential gene expression and chromatin accessibility were estimated using generalized linear models as described in the **Methods** section. In these negative binomial models (transcript and peak count data), observations with more variance (e.g. low counts) contribute less to the coefficient estimates. The presented results therefore represent weighted common features across the three cell lines. We added statements to this effect in the **Methods** section (revised manuscript line 396-397 and 420-421).

7. **Please check for correctness by referencing “Figure 4e” (page 7, line 264) instead of Figure 4c. Also in Figure 4c, check for the correctness of FDR values of scatter plot. Figure 4e lacks ordinate axis units and title.**

We apologize for the oversight. The correct reference is **Figure 3e** (middle panel), the progression-free survival in IFM 2009, where all patients received VRd therapy per study protocol. The FDR values shown in **Figure 4c** are correct. The unadjusted p-values were close to 1.0, after adjusting for the large number of comparisons, the FDR is 1.0 for all individual comparisons. The missing ordinate axis label in **Figure 4e** has been added. The manuscript has been revised to reflect these changes (revised manuscript line 221 and 829-830).

8. **Please check for assay correctness between the two sentences on page 9, line 312.**

The sentence has been revised for clarity and an additional reference has been added (revised manuscript line 261-264).

9. **Minor errors: page 4, lines 116-117; page 26, line 942; page 28, line 981.**

Thank you for pointing out these typographical errors. The manuscript has been revised accordingly (revised manuscript line 103-104, line 764-765, and line 804-806).

Response to Comments by Reviewer #2

1. **The additional scRNA/ATAC –seq data clarifies the fact that certain differential accessibility regions correlate to differential expression in single cells, demonstrating that the effect is not a bulk effect and that the opening/closing of chromatin and the increased/decreased expression of the PUTATIVE target genes happens indeed in the same cell. These experiments do not validate the CRE-gene targets pairs, as it has not been demonstrated that such region is actually regulatory, or the direct link with this regulatory region and the predicted/assigned gene target. I understand that the experiments to demonstrate the functionality of the regulatory region go beyond the scope of this study, but I would call this are ‘potential’ CREs-gene targets.**

We agree with this distinction, the manuscript has been revised to reflect the fact that these are “potential” target genes that were “predicted” (revised manuscript line 119, 122, 145, 156-157, 173, 748, 782, and 922-924). The figure labels have been revised accordingly.

2. The authors addressed this point in a different way than I intended. What I wanted to be clarified was: Are the DE genes (68 or some of the 10 selected) found in this study also found DE in previous studies? If so, this is an independent validation of the experimental approach and I some cases with the use of different cell liens in presence of stroma.

We have compared the gene signatures from both McMillin et al and Dziadowicz et al and have 6 of 10 (*AKAP12*, *ARAP3*, *FSTL1*, *GADD45A*, *IL6*, *VCAN*) and 11 of 68 genes common between these signatures and our genes. So our data is just a partial validation of the 10 selected targets. Besides differences in cell lines, the incubation times (24 vs 36 hours), cell purification methods (flow vs. magnetic beads), and antibody used (CD138+ versus CD19- / CD38+) as well as differences in analytical approaches may account for some of the differences observed. We have now revised the Discussion section to comment on these findings (revised manuscript line 303-308).

3. **Minor/format: I would suggest the authors to revise the colour schemes as the same colours (blue/pink-red) are used to indicate:**

- +/- stroma (treatment);
- DE and DA (up/down)-gene expression
- Prognostic significance and ASI-/+

As this can be confused with the fact that + stroma also implied increase accessibility and increase gene expression, and it is clear that the treatment induce both changes (Da and DE).

We agree and have revised the color coding for the figures to the following system:

1. **Stroma-**  and **Stroma+** 2. **Decreased DE / DA**  and **Increased DE / DA** 3. **ASI-**  and **ASI+** 
Please see the revised **Figure 1** and **2** (revised manuscript line 733-734and 758-759).

Reviewers' Comments:

Reviewer #1:

Remarks to the Author:

It is considered that the authors have addressed the comments in the previous version of the manuscript NCOMMS-23-00840B.

Nevertheless, please double-check for correctness in these lines of the text:

- lines 72 – 77: essay flow (sentence too long; also check for meaning); should "bone marrow mesenchymal stromal cells" be referred to by their acronym "BMSCs"?; "in" is repeated
- please check for appropriateness of "BMSC" or "BMSCs" along the text (e.g., line 88, line 97, line 100, line 103...)
- line 775: "demonstrating differentially expression of genes"
- line 784: "4626 predicted target genes predicted too"

Reviewer #2:

Remarks to the Author:

I can confirm that the authors addressed all the points highlighted in the revised article. I am happy for the article to be accepted in the current form

Reviewer #1 (Remarks to the Author):

It is considered that the authors have addressed the comments in the previous version of the manuscript NCOMMS-23-00840B.

Nevertheless, please double-check for correctness in these lines of the text:
- lines 72 – 77: essay flow (sentence too long; also check for meaning); should "bone marrow mesenchymal stromal cells" be referred to by their acronym "BMSCs"?

This sentence has been revised for brevity and clarity (*revised manuscript lines 74-76*).

"in" is repeated

Duplicated word removed (*revised manuscript line 75*).

- please check for appropriateness of "BMSC" or "BMSCs" along the text (e.g., line 88, line 97, line 100, line 103...)

Missing plural (BMSCs) added as appropriate (*revised manuscript lines 95, 104, 107, 110, and 257*).

- line 775: "demonstrating differentially expression of genes"

This sentence has been revised (*revised manuscript lines 778-779*).

- line 784: "4626 predicted target genes predicted too"

This sentence has been revised (*revised manuscript lines 787-788*).

In the "Material and Methods" section of the manuscript, authors state that "R and Stata code to calculate the \sum ASI classifier and pertinent survival analyses are provided" (line 553).

However, I have not been able to find this code in the "Code and submission checklist" nor the "Reporting summary" documents. Indeed, in the "Code and software submission checklist", "Additional information", authors state "N/A" when asked to provide a link to the code in an open source repository when available.

All previous submissions included an archive named "R_and_Stata_Code.tar.gz" containing the code in R and Stata as well as example data for the three cohorts and appropriate documentation for use:

- ASI_R.R
- ASI_Stata.do
- GSE24080.txt
- IFM_2009.txt
- MMRF_IA16.txt
- README.txt

We plan to make this archive available as supplementary along with the article. For the purpose of this review, the archive can also be accessed here: <https://github.com/bindermo/mmasi>.

Reviewer #2 (Remarks to the Author):

I can confirm that the authors addressed all the points highlighted in the revised article. I am happy for the article to be accepted in the current form.

Thank you very much for the constructive feedback on this manuscript.